



# Replacing the AMOR by the miniDOAS in the ammonia monitoring network in the Netherlands

A.J.C. (Stijn) Berkhout[1], Daan P.J. Swart[1], Hester Volten[1], Lou F.L. Gast[1], Marty Haaima[1], Hans Verboom[1*], Guus Stefess[1], Theo Hafkenscheid[1], Ronald Hoogerbrugge[1]

[1]National Institute for Public Health and the Environment (RIVM), P.O. Box 1, 3720 BA Bilthoven, the Netherlands
[*]Now at Royal Netherlands Meteorological Institute (KNMI), P.O. Box 201, 3730 AE De Bilt, the Netherlands

*Correspondence to*: A.J.C. Berkhout (stijn.berkhout@rivm.nl)

**Abstract.** In this paper we present the continued development of the miniDOAS, an active differential optical absorption spectroscopy (DOAS) instrument to measure ammonia concentrations in ambient air. The miniDOAS has been adapted for

use in the Dutch National Air Quality Monitoring Network. The miniDOAS replaces the life-expired continuous-flow denuder ammonia monitor (AMOR). From September 2014 to December 2015, both instruments measured in parallel before the change from AMOR to miniDOAS was made. The instruments were deployed on six monitoring stations throughout the Netherlands. We report on the results of this intercomparison.

Both instruments show a good uptime of ca. 90%, adequate for an automatic monitoring network. Although both instruments

produce minute values of ammonia concentrations, a direct comparison on short timescales such as minutes or hours does not give meaningful results, because the AMOR response to changing ammonia concentrations is slow. Comparisons between daily and monthly values show a good agreement. For monthly averages, we find a small average offset of $0.65 \pm 0.28$ µg m$^{-3}$ and a slope of $1.034 \pm 0.028$, with the miniDOAS measuring slightly higher than the AMOR. The fast time resolution of the miniDOAS makes the instrument not only suitable for monitoring but also for process studies.

**1   Introduction**

Ammonia in the ambient air plays an important role in the formation of aerosols (Asman, 1998), through reactions with nitrogen oxides or sulfur dioxide. These aerosols contribute to the total burden of particulate matter and may lead to public health effects (Fischer et al., 2015). When deposited on nature areas, ammonia causes acidification and eutrophication, leading to a loss of biodiversity. Intensive animal husbandry leads to high ammonia emissions in the Netherlands. Hence, it

is of great importance to have a correct understanding of the sources and sinks of ammonia in the Netherlands and of the processes that determine ammonia emissions to the air, its transport, its atmospheric chemistry and its deposition. Therefore, an ammonia monitoring network has been in operation in the Netherlands since 1992, providing hourly concentration measurements, currently at six locations. These measurements are part of the Dutch National Air Quality Monitoring Network.





This is one of the very few ambient ammonia monitoring networks with high temporal resolution in the world. Elsewhere, ammonia monitoring takes place with diffusion tubes, with data coming in every week, every few weeks or even every month. The hourly values measured in the National Air Quality Monitoring Network thus constitute a unique dataset going back for more than two decades (van Zanten et al., 2017).

The monitoring data enables analyses of trends and processes determining the temporal variation in the concentrations. It is also used for the calibration of the Measuring Ammonia in Nature (MAN) network in the Netherlands (Lolkema et al., 2015). In recent years, extension of the MAN enabled the use of its data for the trend analyses, supplementing the data gathered from the monitoring network. That network could therefore be reduced from the original eight to the current six operational stations.

From its inception in 1992, the ammonia monitoring network used the AMOR instrument (Buijsman et al., 1998). This is a wet denuder instrument, based on chemical absorption. The complexity of the instrument, combined with the need to exchange spent chemical solutions, made the instrument expensive to operate and maintain. Moreover, after more than twenty years, the instruments used were life-expired. For these reasons, we started searching for a replacement instrument. The literature, e.g. Schwab et al. (2007) or Von Bobrutzki et al. (2010), did not give a clear-cut advise. We conducted

intercomparison campaigns in 2007-2008 and 2009-2010. From these campaigns, it became clear that the best instrument would be an active DOAS instrument (Differential Optical Absorption Spectroscopy). All other instruments tested used inlet lines to sample air, leading to ammonia sticking to inlets, filters and instrument surfaces, interacting with water or emanating from trapped ammonia aerosols (Parrish and Fehsenfeld, 2000). This results in delay effects, a reduced temporal resolution and sensitivity to interference from aerosols. Active DOAS, with its open optical path, avoids all these problems completely.

The DOAS technique was first developed in the 1970s (Platt et al., 1979). Since then, it has been used for measuring a multitude of atmospheric constituents (Platt and Stutz, 2008), e.g. $NO_2$, $SO_2$ (Avino and Manigrasso, 2008), mercury (Edner et al., 1986) and aromatic compounds such as benzene (Barrefors, 1996).

To the best of our knowledge, the first mention of an active DOAS for ammonia in the 200-230 nm wavelength range was by Edner et al. (1990). They used their instrument to measure ambient ammonia concentrations in rural and urban areas (Edner

et al., 1993; Gall et al., 1991). In 2002, Mount and co-workers reported on the construction of a similar system (Mount et al., 2002). They used it to measure ammonia emissions from local sources at a research dairy (Rumburg et al., 2006; Rumburg et al., 2004). Around that time, our group started working on an ammonia DOAS by modifying a commercial instrument, a DOAS 2000 from Thermo (now discontinued). We used the system for concentration and deposition measurements (Volten et al., 2012b). Our modified version showed much improvement over the original; it is described in (Volten et al., 2012a).

As mentioned above, from the modified DOAS we developed the miniDOAS, also described in (Volten et al., 2012a). That system inspired Sintermann and co-workers to build their own, modified, miniDOAS system (Sintermann et al., 2016). They measured artificial sources, concentration differences due to grazing cattle, and the results of manure application to fields. An overview of these systems is given in Table 1.





The table shows that instrument performance has increased over the years: detection limit and integration time have decreased simultaneously. The other entries in the table show how the groups tackled various challenges posed by the DOAS technique. The line "set-up" in the table describes the instrument lay-out. The Edner instrument is bistatic, i.e. light source and detector are in two separate locations. This makes alignment difficult, as both parts of the set-up must be precisely

aligned. It also requires power to be available at both ends of the path. For this reason, all other instruments have light source and detector combined in a single instrument. Such a monostatic set-up folds the path back with a retroreflector. Losses due to the extra reflections at the reflector and to the optical geometry (part of the light beam gets reflected back into the light source, missing the detector) makes this set-up less light efficient. This is reflected in the shorter path lengths used.

In 1990, Edner and co-workers selected a scanning slit monochromator with a photomultiplier tube as detector, noting that a

spectrograph with an array detector had many advantages but was financially out of reach. Improvements in semiconductor technology made diode arrays and CCDs accessible to the makers of the other instruments. The miniDOAS uses an uncooled CCD rather than a cooled detector as the other instruments do, sacrificing some performance for lower costs. To help tackling stray light, the Edner, RIVM and miniDOAS instruments use an interference filter to block out visible light. The Mount system instead employs a double spectrograph, eliminating the need for an interference filter. The Sintermann system

uses a deuterium arc lamp rather than a xenon arc lamp, as the other systems do. A deuterium arc lamp emits hardly any visible light, which is the reason this system can do without an interference filter. Its lack of visible light also makes this system less obtrusive, which may facilitate its placement and avoids attracting vandalism. A disadvantage of a deuterium arc lamp is its shorter lifetime when compared to a xenon arc lamp. This tipped the balance towards xenon arc lamps for the other systems.

Another consideration is that its copious amounts of visible light makes a xenon arc lamp inherently more eye-safe than a deuterium arc lamp, as the natural reaction of people to the bright visible glare of the xenon arc lamp is to look away. The pale purple glow of the deuterium arc lamp offers no such reflex, so onlookers may inadvertently be exposed to ultraviolet radiation. For an instrument in a monitoring network, that is to operate unattended 24 hours per day, this extra safety offered by the xenon arc lamp is an important advantage.

When comparing the miniDOAS with the other systems, we note that the detection limit is slightly higher than of the two contemporary systems, but lower than of the systems built 10 and 22 years earlier. The path length of the miniDOAS is shorter than that of all other systems, reflecting the low-power lamp used. The use of an uncooled CCD does not affect the performance unduly.

The miniDOAS shows adequate performance for a monitoring network: a detection limit of 0.25 µg m$^{-3}$, an accuracy of 0.25

µg m$^{-3}$, a true time resolution of 1 minute and an instrument uptime exceeding 90%. When compared to the AMOR, purchasing price and maintenance requirements are much lower, leading to an attractive cost reduction while increasing measurement quality. Before the transition from AMOR to miniDOAS was made on 1 January 2016, an extensive comparison period was conducted, from September 2014 to December 2015, in which both instruments were operated in





parallel. This paper describes the implementation and performance of the miniDOAS, the intercomparison with the AMOR and some issues associated with the transition.

## 2 Measurement methods

### 2.1 The Dutch National Air Quality Monitoring Network

The Dutch National Air Quality Monitoring Network (LML) was established more than 50 years ago, to monitor air pollution. Starting in 1992, ammonia has been measured at eight of these stations (since 2014 on six stations). The instrument used from 1992 to 2016 was the AMOR (Ammonia Monitor), see Sect. 2.2.1. A map of the network is shown in Fig. 1. More detailed maps of the monitoring sites are shown in Appendix A.

### 2.2 The AMOR

#### 2.2.1 Description of the instrument

The AMOR, Ammonia MonitoR, is an automatically operating continuous-flow denuder system. It was developed at ECN in the early 1990s from the AMANDA (Wyers et al., 1993). The AMOR differs from the AMANDA mainly in its remote control options and its ability to operate unattended for prolonged periods of time (up to 4 weeks). It is described in detail in Erisman et al. (2001), Wyers et al. (1993) and in Buijsman et al. (1998). The procedure in which this instrument was selected

for use in the monitoring network LML, as well as tests of its performance and the results of the first years of measurements are described in Buijsman et al. (1998) and in Mennen et al. (1996). The published specifications for the AMOR were a detection limit of $0.01\ \mu g\ m^{-3}$, an accuracy of 2% and a time resolution of 3 min (Erisman et al., 2001).

#### 2.2.2 Implementation in the monitoring network LML

The instrument was installed inside the climate-controlled housing of the monitoring stations. Air was sampled from an inlet

on the roof of the housing, at 3.5 m above ground level. The air flow through this inlet was $250\ m^3\ h^{-1}$. From this air flow, a small fraction ($25\ L\ min^{-1}$, or 0.6%) was sampled, just after a 90° turn in the inlet tube, and fed into the AMOR. This arrangement served to minimise the amount of particulate matter being sampled by the AMOR.

#### 2.2.3 Calibration

Calibration of any trace gas monitor is preferably done by offering the instrument a gas stream with a high but realistic

concentration of the gas to be measured. For ammonia, this means a concentration of e.g. $400\ \mu g\ m^{-3}$. A gas bottle with a mixture of such an ammonia concentration in e.g. nitrogen is not stable. For the AMOR in the monitoring network, calibration was therefore carried out by offering the instrument a solution of $NH_4^+$ of a known concentration, corresponding





to a realistic ammonia concentration in the ambient air. Calibration of the system took place automatically, every 80 hours, with $NH_4^+$ solutions in two concentrations.

### 2.2.4    AMOR dataset used in this study

The AMOR data were corrected for a small offset compared to data published in national and international databases. The
offset is caused by a digital to analogue conversion of the data. See Appendix B for more information.

## 2.3    The miniDOAS

### 2.3.1    Description of the instrument

The miniDOAS is extensively described in Volten et al. (2012a). The instrument uses a xenon lamp as ultraviolet light source and a retroreflector to measure optical absorption spectra along an open path, typically 42 m long. It uses the DOAS
technique, Differential Optical Absorption Spectroscopy, to retrieve concentrations of several atmospheric trace gases along this path. See Fig. 2 for a schematic representation of the optical set-up of the instrument.

### 2.3.2    Retrieval of concentrations

The spectral window used is from about 203.6 to about 230.9 nm, the precise window differs slightly between instruments. In this region, three gases commonly present in the atmosphere show specific absorption patterns: $NH_3$, $SO_2$ and NO. These
three gases are retrieved together. Other atmospheric constituents either do not absorb in this region (e.g. $NO_2$) or have absorption features that change only slowly with the wavelength (e.g. $O_3$), those are filtered out by the fitting algorithm.

Central to the DOAS technique is the Lambert-Beer law. Eq. (1) is one form to write it (CEN, 2013):

$$I_{meas}(\lambda) = I_0(\lambda) \cdot e^{(-a(\lambda) \cdot c \cdot l)} \tag{1}$$

Here, $I_{meas}(\lambda)$ is the measured spectrum, $I_0(\lambda)$ is the intensity spectrum as emitted by the instrument, $a(\lambda)$ is the specific
absorption coefficient of the component through which the light passes, $c$ is the concentration of that component and $l$ is the optical path length. In the open atmosphere, light is attenuated not just by absorption but also by Rayleigh and Mie scattering. The wavelength dependent attenuation by the optical system must also be taken into account. The key to the DOAS technique is to separate narrow-band absorption features in the specific absorption spectrum $a(\lambda)$ from broadband features due to interfering compounds, atmospheric scattering and the intensity spectrum of the light source. This leads to the
following equation:

$$I_{bgc}(\lambda) = I_0'(\lambda) \cdot e^{\Sigma(-a_i'(\lambda) \cdot c_i \cdot l)} \tag{2}$$

Here, $I_{bgc}(\lambda)$ is the background corrected measured spectrum, see Sect. 2.3.4 for the determination of this background. $I_0'(\lambda)$ is the differential initial intensity, all broadband features are collected in this term. $a_i'(\lambda)$ is, for component $i$ to be measured, the part of the specific absorption spectrum containing the narrow-band absorption features. In logarithmic form:





$$\ln\left(\frac{I_{bgc}(\lambda)}{I_0'(\lambda)}\right) = \Sigma(-a_i'(\lambda) \cdot c_i \cdot l) \tag{3}$$

To approximate $I_0'(\lambda)$, we use a moving average of the measured spectrum, denoted as $[I_{bgc}(\lambda)]_{\text{moving average}}$ in Eq. (4) below. We found two consecutive passes with averaging over 41 channels each time to work well. We define the DOAS curve $DC(\lambda)$:

$$DC(\lambda) = \ln\left(\frac{I_{bgc}(\lambda)}{[I_{bgc}(\lambda)]_{\text{moving average}}}\right) \tag{4}$$

To retrieve the concentrations, we use a three component least-squares fit (Kendall and Stuart, 1976; Volten et al., 2012a):

$$\sigma^2 = \sum_{j=1}^{n} \frac{(DC(\lambda)_j - \alpha \cdot X(\lambda)_j - \beta \cdot Y(\lambda)_j - \gamma \cdot Z(\lambda)_j)^2}{(n-3)} \tag{5}$$

Here, $\sigma$ is the standard deviation of the fit. The wavelength is denoted by $j$, $n$ is the total number of wavelengths, $X(\lambda)_j$, $Y(\lambda)_j$ and $Z(\lambda)_j$ are reference spectra for $NH_3$, $SO_2$ and $NO$, respectively. The parameters $\alpha$, $\beta$ and $\gamma$ are proportional to the

concentrations of $NH_3$, $SO_2$ and $NO$ to be retrieved. The retrieval algorithm minimises $\sigma^2$ to find the best fit. The concentration of $NH_3$ is calculated according to Eq. (6):

$$c_{NH_3} = \frac{\alpha}{l} \tag{6}$$

Concentrations for $SO_2$ and $NO$ are calculated analogously.

### 2.3.3 Calibration: span measurements

To determine the reference spectra $X(\lambda)$, $Y(\lambda)$ and $Z(\lambda)$ used in Eq. (5), two methods exist (CEN, 2013): calibration with complete spectral modelling using reference spectra, and gas cell calibration with and without including the atmosphere. Both methods have their advantages and difficulties.

The spectral modelling method involves modelling the complete system. This requires knowledge of the instrument line shape function (ILS) of the spectrometer, of the presence of straylight in the spectrum, of the linearity and dark current of the

detector, and of the differential absorption coefficient of each component at the wavelengths used. The last parameter can be obtained by measuring a spectrum with the instrument itself, or by convolving a high-resolution absorption spectrum from the literature with the ILS.

For gas cell calibration, a cell with the gas for which the reference spectrum is to be determined is placed in the light path. The concentration of the gas should be known. Care should be taken that the gas is stable in the cell, or a flow-through cell

should be used. When applying this method with inclusion of the atmosphere, only an incremental calibration can be performed, i.e. the system will measure an increase due to the gas in the gas cell with respect to the atmospheric background concentration. This requires stable atmospheric conditions.

For the gas cell method with exclusion of the atmosphere, the light path should be routed directly from the light source through the gas cell into the detector. This eliminates all atmospheric influence. It also allows a calibration under zero gas

conditions.





The systems discussed above all use the gas cell method to calibrate. Edner et al. (1993) do not specify whether they included the atmosphere or not. Mount et al. (2002) discuss the possibility of using the spectral modelling method but prefer the gas cell method, because it automatically convolves the ILS with the gas cross section spectrum. They did exclude the atmosphere. Sintermann et al. (2016) do not specify whether they included the atmosphere or not. They did apply the

modelling method, as a check on their calibration spectrum, for $NH_3$ only. For the RIVM DOAS, we used the gas cell method, with exclusion of the atmosphere (Volten et al., 2012a).

For the miniDOAS, we use this method as well. We place a 75 mm path length quartz flow cell in the instrument, see Fig. 2. Pressure and temperature of the gas are continuously measured so that the amount of gas in the cell is known. Because the path in the optical cell is, at 75 mm, only 1/560$^{st}$ of the full 2 x 21 m path that is used in the open air, the concentration in the

cell must be 560 times the ambient concentration. This is an important advantage, as it enables DOAS systems to be calibrated with high-concentration gas mixtures. These are much more stable than the mixtures at ambient concentrations that an air-sampling instrument would use. The gas mixtures used are listed in Table 2.

To exclude the atmosphere, we originally used a shortened optical path of 1 m rather than 42 m. However, we found that, for the miniDOAS, this short path yielded distorted spectra compared to spectra measured over a full length optical path. We

attribute this distortion to the difference in Rayleigh scattering over a longer versus a shorter path. At the short wavelengths we use for ammonia DOAS, this effect is much more pronounced than in DOAS applications at longer wavelengths, since the intensity of Rayleigh scattering is proportional to the inverse fourth power of the wavelength of the light. In addition, the tail from the Schumann-Runge absorption bands of $O_2$ affects the spectrum at the low wavelength side (Yoshino et al., 1984). The combined distortion negatively affected the fitting procedure. Therefore, we decided to measure the reference

spectra with the full optical path. A disadvantage of this long path is that any gas present in the atmosphere will leave its spectral signature on the reference spectra, whereas those spectra are assumed to contain only the known concentration of the target gas. To address this issue, we set up the miniDOAS that has its reference spectra measured (the miniDOAS under test) in the laboratory next to another DOAS (the reference DOAS), with the optical paths of both instruments running parallel, so that they measure the same parcel of ambient air. Both instruments measure a full-length outdoor atmospheric path.

The reference DOAS can be any previously calibrated DOAS, or indeed, any instrument capable of measuring ammonia. It turned out to be most convenient to use an RIVM DOAS (as described in Volten et al. (2012a); see also Table 1) that had been calibrated with exclusion of the atmosphere. This DOAS system reports values at 5 minute intervals, it has a detection limit of 0.15 μg m$^{-3}$ for $NH_3$ and it receives regular maintenance.

We use the reference DOAS to determine the concentrations of $NH_3$, $SO_2$ and NO in the ambient air during the reference

spectra measurements. If those ambient concentrations are too high (>10 μg m$^{-3}$ for $NH_3$, >5 μg m$^{-3}$ for $SO_2$ or >20 μg m$^{-3}$ for NO) the resulting reference spectra are rejected and measured anew. In this way, we make sure any remaining effects are small:





- The concentrations in the gas cell are chosen so that the equivalent concentrations in the atmosphere are realistic, but high. As an example, for $NH_3$, an equivalent concentration of 500 μg m$^{-3}$ is used. If $NH_3$ is present in the outdoor optical path at the typical ambient concentration in the Netherlands of 5 μg m$^{-3}$, the resulting error is 1%.

- The effects of one gas being present in the reference spectrum of another gas is also small. As an example, suppose some ambient NO is present while the reference spectrum for $NH_3$ is being measured. The resulting spectrum will have the spectral features of both $NH_3$ and NO, but since $NH_3$ is present in a much higher concentration, the features of NO will be negligible.

As measuring $NH_3$ concentrations is the prime aim of this instrument, the calibration for $NH_3$ is checked with a Primary Reference Material (PRM), obtained from VSL in the Netherlands. This check is done by measuring the apparent concentration when a PRM gas mixture of $NH_3$ in $N_2$ is passed through the flow-through calibration cell, using the same procedure as outlined above for the measurement of the reference spectra. If the concentration reported by the instrument is within the tolerance of the PRM gas, as indicated by its manufacturer (this amounted to 2%, see Table 2), the instrument is considered to have passed the test. If the reported concentration is outside the tolerance, the instrument is considered to have failed the test. In the latter case, its reference spectra are discarded and measured anew.

Reference spectra are measured in the following cases:

- Before a new system is deployed for the first time.
- After a system has its xenon lamp replaced. This usually takes place after operating a system for one year.
- After any repairs to the spectrograph.

Because replacing the xenon lamp makes measuring new reference spectra necessary, such a replacement takes place by exchanging a system with a life-expired lamp with a system with a new lamp installed and newly measured reference spectra present. This minimises the instrument downtime.

### 2.3.4 Calibration: zero measurements and dark current corrections

The background subtracted spectrum referred to in Eq. (7) is calculated as follows:

$$I_{bgc}(\lambda) = \frac{I_{meas}(\lambda) - I_{dark}(\lambda, T)}{I_{background}(\lambda)} \tag{7}$$

Here, $I_{bgc}(\lambda)$ is the background corrected spectrum, $I_{meas}(\lambda)$ is the spectrum measured by the spectrograph, $I_{dark}(\lambda, T)$ is the temperature-dependent dark spectrum and $I_{background}(\lambda)$ is a reference spectrum containing no spectral features from atmospheric gases.

The temperature-dependent dark spectrum $I_{dark}(\lambda, T)$ is determined by placing the spectrograph in a dark temperature-controlled room and measuring the spectral response at a number of temperatures and integration times.

Dividing by the reference spectrum $I_{background}(\lambda)$ improves the signal-to-noise ratio of the measurement. To measure $I_{background}(\lambda)$, we need an optical path that is free of $NH_3$, $SO_2$ and NO. For the RIVM DOAS reported on in Volten et al. (2012a), we achieved this by shortening the optical path, thereby excluding the ambient air with its background





concentrations. Other groups handled this in a similar way, e.g. Mount et al. (2002), who used a shortened path away from local sources.

An alternative is to somehow deal with the background concentrations of $NH_3$, $SO_2$ and NO along the full length optical path. Sintermann et al. (2016) identify three ways to do this:

1) Monitor for an extended period of time. During this time, some episodes with near-zero ambient concentrations are likely to occur.

2) Measure $I_{background}(\lambda)$ at a remote location where ambient concentrations are assumed to be very low. Alternatively, create an artificial low-concentration environment in the laboratory, over the full optical path.

3) Remove traces of $NH_3$, $SO_2$ and NO by excluding narrowband absorption from $I_{meas}(\lambda)$. This may be done by applying a
low-pass filter.

They tested method 3) and found it to work well; however, the method also introduced extra uncertainty to the results.

We used method 1) for the reference DOAS mentioned above. A disadvantage of this method is that $I_{bgc}(\lambda)$ is only available after operating the instrument for a long time, of many weeks or even months. This is fine if measurement results are analysed after a monitoring period, but it is not acceptable in a monitoring network, when measurements are to be reported in
near real-time. For the miniDOAS, we adopted a different approach.

We use the set-up with a reference DOAS and the miniDOAS under test running parallel that we described in Sect. 2.3.3. The $I_{background}(\lambda)$ of this reference DOAS was measured using method 1), as outlined above. The reference DOAS indicates when the interfering gases reach low concentrations in the ambient air. Spectra measured with the miniDOAS under test during those episodes are used as its $I_{background}(\lambda)$. This $I_{background}(\lambda)$ still contains the signature of low ambient concentrations
of $NH_3$, $SO_2$ and/or NO, typically below 5 µg m$^{-3}$ for $NH_3$. We determine the average differences in concentration between miniDOAS and reference DOAS and correct for these in the retrieval procedure. For an example, see Fig. 3.

### 2.3.5 Error sources, detection limit and precision

We distinguish between random and systematic error sources. The random error sources, e.g. the correction for the dark current, end up in the residual spectrum. Their combined magnitude is estimated by the standard error that is reported by the
fitting algorithm (Stutz and Platt, 1996). To use the standard error to estimate upper limits for the detection limit and the precision we select episodes at the monitoring stations with very low ambient concentrations. An example of such an episode is shown in Fig. 4. Note that this episode still contains both a background concentration and some natural variability in the ammonia concentration. From such episodes, we estimate the upper limit of the precision to be 0.25 µg m$^{-3}$ and the detection limit to be 0.25 µg m$^{-3}$, both at the instrument time resolution of 1 minute (Table 3). The precision of hourly averaged data
we estimate to be 0.1 µg m$^{-3}$.





The systematic error sources are listed in Table 4. As an example, the calibration gas concentration is known with a precision of 2%. This concentration is used only once, in the preparation of the reference spectra, and the associated error is the same for as long as this spectrum is used. These will never be estimated by the fitting algorithm.

We use a tape measure to determine the path length, we estimate that we do this with a precision of 5 cm, or 0.25% of 20 m.

The calibration gas concentration is given by the manufacturer with a precision of 2%. The zero concentration measured by the reference DOAS we determine with a precision of 0.45 $\mu$g m$^{-3}$, i.e. 3 times its precision of 0.15 $\mu$g m$^{-3}$ (Volten et al., 2012a). Combined, we estimate the detection limit to be 0.45 $\mu$g m$^{-3}$ and the precision to be 2.25%, with a minimum of 0.25 $\mu$g m$^{-3}$.

### 2.4    Instrument intercomparison campaign

To fully characterise the differences between the AMOR and the miniDOAS, an instrument intercomparison campaign was conducted. From 2 September 2014 to 31 December 2015, on each of the six operational stations, both an AMOR and a miniDOAS were operated in parallel. Both instruments were operated in the regular monitoring network mode. For the miniDOAS, operating procedures and the data transfer set-up were updated and refined during the campaign. This did not influence the measurement data, as these refinements dealt with issues not directly affecting the measurements or the
retrieval. These changes in the procedures did improve the instrument uptime.

The AMOR measurements were conducted under ISO 17025 accreditation.

#### 2.4.1    Height difference assessment

AMOR measurements have always been conducted with an air inlet at 3.5 m above the local ground level. This is the standard air inlet height for the Dutch Air Quality Monitoring Network. For the miniDOAS this height was considered
unpractical, as it would mean mounting the instrument outside the station housing, or using a complex optical setup. Instead, a measuring altitude was chosen of 2.2 m. This corresponds to the highest practical mounting position inside the station housing.

The effect of the difference in measurement height was studied using passive samplers. At all stations, three sets of three Gradko passive sampler tubes (Lolkema et al., 2015) were installed:

-   Set 1: attached to the AMOR air inlet at 3.5 m height, above the station housing roof.

-   Set 2: at 3.5 m height (the AMOR air inlet height) in a separate mast, halfway the miniDOAS optical path.

-   Set 3: in the same mast, but at 2.2 m height (the height of the miniDOAS optical path).

See Fig. 5 for an overview of the situation.

The passive sampler sets were exchanged and analysed monthly. Measurements took place between January and December
2014. Table 5 lists in which months and on which monitoring stations the samplers were deployed.





## 3    Results

### 3.1    Dataset and uptimes of the miniDOAS and AMOR

In Fig. 6 uptimes of the AMOR and miniDOAS systems are given over the period from 1 September 2014 to 1 September 2015. We excluded periods when the miniDOAS systems were in the laboratory for instrument characterisation, when
station housings were renewed, et cetera.

Uptimes for both the AMOR and the miniDOAS instruments were mostly between 80-90%, as is required for instruments in a monitoring network. Note that the miniDOAS systems were during this period not formally part of the monitoring network and therefore not under continuous surveillance, in contrast to the AMOR instruments.

Comparing any two instruments can only be done with data gathered on a timescale that permits both instruments to produce
meaningful data. Both AMOR and miniDOAS generate a data point every minute. However, it takes time - about half an hour - for ammonia to be processed by the AMOR system. This causes a delay and a smoothing effect in the AMOR values. In addition, instruments that employ inlet lines are known to suffer from a memory effect due to ammonia sticking to walls of the inlet line (Parrish and Fehsenfeld, 2000). This ammonia may be released later, depending on temperature and relative humidity. The miniDOAS has no inlet lines and shows an instant response to ammonia in the air. Therefore, comparison
between AMOR and miniDOAS minute value is not feasible, as will be shown below. Comparison between hourly values is complicated but possible, comparison between daily and monthly values works fine.

We will briefly illustrate the smoothing and delaying effects of the internal works and inlet lines of the AMOR on its data by applying a similar effect to the miniDOAS data using the simple formula in Eq. (8) (Volten et al., 2012a; Von Bobrutzki et al., 2010):

$$c'(t) = f\, c(t) + (1 - f\,)c'(t - 1) \qquad (8)$$

where $c'(t)$ is the delayed smoothed concentration, $c(t)$ is the measured miniDOAS concentration data and $f$ is a smoothing factor which would be unity for an instrument equally fast as the miniDOAS. We use an e-folding time $\tau_{1/e}$ of 1 hour, where $\tau_{1/e} = 1/f$. The value of 1 hour was adopted for illustration purposes, similar to what was reported earlier in Volten et al. (2012a). After applying the smoothing and delaying effect the miniDOAS data is remarkably similar to the AMOR data, as
illustrated in Fig. 7. It is not our aim to find the perfect smoothing and delaying curve for the miniDOAS data to reproduce the AMOR data. We just wish to illustrate that for comparisons of the miniDOAS and the AMOR data it is more meaningful to compare averages over longer time intervals. Below we give some examples of the AMOR and miniDOAS data compared for different, increasing time intervals, starting with hourly values, to daily values to monthly values.

### 3.1.1    Hourly values compared

When comparing hourly values of the miniDOAS and the AMOR the delay effect is less pronounced than for the comparison of the minute values, but a smoothing effect and a delay is still clearly visible as illustrated in Fig. 8, containing hourly data for a selected period at the monitoring station in Vredepeel. This station has strongly varying concentrations with



relatively high ammonia peaks, up to several hundred micrograms per cubic meter. Since the miniDOAS suffers no smoothing effect, it captures the concentration peaks more effectively than the AMOR.

The delay effect is clearly visible during 17 March. In a period when the concentrations are less variable and less extreme, e.g. from 19 to 22 March, the AMOR and miniDOAS data agree much better. This is reflected in the scatter plot shown in Fig. 9. Here, all the largest deviations from the y = x line are cases of the miniDOAS value being larger than the AMOR value.

On a monitoring station where on average the ammonia concentrations are much lower, such as De Zilk, the delay and smoothing effects are visible as well. As is apparent from the scatter plot (Fig. 10), deviations from the y = x line still occur, again the deviations are larger on the side where the miniDOAS values are higher than the AMOR values.

### 3.1.2 Daily values compared

Figure 11 shows daily averages of ammonia concentrations measured by the AMOR and miniDOAS systems in Vredepeel in the period from 1 September 2014 up to and including 31 December 2015. For each instrument, only days on which at least 18 hourly values were measured (75% uptime) were included. Here the comparison between the two measurement methods is quite good despite the large dynamics in the concentrations, demonstrating that averaging over a day effectively removes the main differences between the data of both instruments.

This is also apparent when we compare the scatter plots of the hourly values at Vredepeel (Fig. 9) with the scatter plot of the daily values at the same station (Fig. 12). The latter shows more clustering around the y = x line, although some extremes remain.

### 3.1.3 Monthly values compared

In Fig. 13 we give monthly averages for both miniDOAS and AMOR. The monthly averages are based on hourly values, but only on those where both AMOR and miniDOAS values were simultaneously available. A monthly average is included when it is based on at least 100 hourly values. The series of monthly values tend to agree well, showing very similar patterns. In many cases, though not all, the AMOR values tend to be slightly below the miniDOAS monthly values. Some episodes, e.g. September 2014 to March 2015 for Wekerom, agree really well. Some other episodes, e.g. July to December 2015 for the same station, show less agreement. The reasons are so far unknown. There is no correlation between high AMOR-miniDOAS differences and episodes with high or low values. Neither is there any seasonal influence.

To evaluate the comparability of the AMOR and miniDOAS data the monthly values in Fig. 13 have been used for a orthogonal regression plot presented in Fig. 14. The number of data pairs included is 89 and the $R^2 = 0.94$.

This orthogonal regression yields a relation between the values of miniDOAS and AMOR given by Eq. (9)

$$miniDOAS = 1.034(28) \cdot AMOR + 0.65(28) \tag{9}$$

where the uncertainty (1σ) in the last two digits of the slope and the offset are given between brackets.





Finally, in Table 6, we list the average concentrations based on pairs of hourly values in the period from September 2014 until the end of December 2015. Standard deviations are given as well, reflecting the variability in the measured concentrations. The annual averages are comparable, but in all cases, the AMOR values are somewhat lower than those of the miniDOAS.

### 3.2    Results of the height difference assessment

Table 7 lists the annual averaged differences in ammonia concentrations measured with passive samplers at different measurement heights averaged over all sites, and standard 2-sigma errors. Given the small concentration differences and the relatively large statistical variance associated with these passive samplers, analysis per station or per season is not feasible with this dataset.

We see no significant difference between the set at the AMOR inlet (at 3.5 m) and the set at the miniDOAS path (at 2.2 m). Results do show a difference between the two measurement points at 3.5 m, i.e. those at the AMOR air inlet and at the separate mast. The concentrations at the AMOR air inlet are lower. This may be explained by the station housing influencing the air flow: the air sampled by the AMOR is not pure air from 3.5 m height, it is mixed with air from lower heights forced upwards by the station housing. In both cases the statistical error is substantial.

### 4    Discussion

Analysing the results obtained in the comparison we see that the uptime of both instruments is comparable. At 80-90%, the miniDOAS uptime is adequate for an instrument in an automated monitoring network. We expect that the uptime of the miniDOAS will further improve in 2016, as from then on the instrument will benefit from the regular monitoring of the network performance.

#### 4.1.1    Timescale of the intercomparison

Both instruments provide minute values of ambient ammonia concentrations. When looking at short timescales (minutes, hours) we see relatively large differences between the datasets. The differences get smaller as the timescale gets longer. When we look at the fits in the scatterplots of hourly, daily and monthly averages (Fig. 9, Fig. 10 and Fig. 14, respectively) we see that the slopes approach unity: 1.54 for hourly averages, 1.27 for daily and 1.03 for monthly averages. The offsets approach zero, from -7.34 µg m$^{-3}$ for hourly averages, via -3.06 µg m$^{-3}$ for daily to 0.65 µg m$^{-3}$ for monthly averages.

For this reason, we focused our comparison on longer timescales: daily and monthly values. Daily value pairs showed good agreement in a direct comparison, i.e. when the concentration values are plotted in the same graph (see e.g. Fig. 11). The smoothing and delay effects that are apparent in the minute and hourly values have largely disappeared. However, scatter plots (see e.g. Fig. 12) show still some deviations from y = x, indicating that some delay effects are still not smoothed out. This is to be expected, a high peak just before the transition to a new day will cause differences in two consecutive days. In



the monthly averages, because there are far fewer transitions, such extremes have disappeared. This makes monthly averages the timescale of choice for the intercomparison.

## 4.2    Intercomparison at longer timescales

Monthly averaged concentrations show a linear relationship, as indicated in the previous paragraph. We conclude that for monthly averages the instruments compare well. Over the whole comparison period there is an average offset of $0.65 \pm 0.28$ µg m$^{-3}$ and a slope of $1.034 \pm 0.028$ between the techniques. Thus, the miniDOAS measures on average slightly higher than the AMOR, over all concentration ranges.

From a scientific point of view this correspondence is excellent, especially since two completely different measurement techniques are used. As a reference, we refer to a study by Von Bobrutzki et al. (2010) that shows much larger discrepancies between different techniques. The systematic difference found between AMOR and miniDOAS amounts to roughly 10% of the typical ammonia background concentrations in the Netherlands of around 5 µg m$^{-3}$. Fortunately, not so much the absolute difference between the techniques is politically relevant but any jumps in ammonia trends, see e.g. (van Zanten et al., 2017; Wichink Kruit et al., 2017) for two studies in which this data is being used. We will discuss some possible explanations for the difference between the techniques.

## 4.3    Possible explanations for the difference between the techniques

### 4.3.1    Possible difference due to height difference

The effect of the difference in measurement altitude (the AMOR measured at 3.5 m, the miniDOAS at 2.2 m) was studied using passive samplers.

The results reported in this paper show no significant difference between AMOR inlet and miniDOAS path, so they offer no explanation for the observed positive bias between miniDOAS and AMOR. The results do show a difference between both measurement points at 3.5 m, indicating that the AMOR measurement may be influenced by the station housing resulting in lower measured values. In both cases the statistical error is substantial, and consequently the results are inconclusive. Further research with more precise equipment would be needed to reduce the statistical error and study the effects of the altitude difference between 2.2 and 3.5 m, and also the possible influence of the station housing.

### 4.3.2    Possible difference due to AMOR validation procedure

AMOR data is validated based on concentration data only. No AMOR instrument parameters are included in the validation procedure. Closer inspection of validated AMOR values show periods after maintenance where values approved in the validation procedure may - in hindsight - be considered too low and erroneous. This conclusion can only be drawn when the miniDOAS dataset is used as an additional validation tool. Removing these data from the comparison (dubious as it would





be from a scientific point of view) would however improve the comparison only slightly. Therefore, the validation procedure can be ruled out as a major source of the offset.

### 4.3.3 Possible difference due to ammonia loss in the AMOR inlet line

It is conceivable that ammonia is lost in the AMOR air inlet system, as this is a known effect in ammonia inlet lines

(Yokelson et al., 2003). However, the AMOR air inlet system has been designed to minimise such effects. Especially the relatively high airflow through the instrument, of 25 L min$^{-1}$ rather than the mL min$^{-1}$ flows found in other instruments, should be effective in minimising these effects. As discussed in Sect. 3, no indication for ammonia loss was found in the measurement data. It seems therefore unlikely that ammonia loss is a major contributor to the bias found.

### 4.3.4 Possible difference due to AMOR calibration procedure

It should be noted that AMOR calibrations are performed using calibration fluids, and thus only pertain to the 'liquid' part of the instrument, after ammonia has been absorbed in the denuder. Any losses in the airborne phase, e.g. in the inlet system, are not included in the calibration procedure. As stated previously, the reason for omitting this part in the calibration procedure is that it is virtually impossible to generate an adequate calibrated gas flow, as the AMOR tries to minimize inlet effects by using a very high airflow of 250 m$^3$ h$^{-1}$, from which a further 25 L min$^{-1}$ is sampled by the instrument. We have

not been able to study this aspect further in the framework of this comparison.

### 4.3.5 Possible difference due to miniDOAS zero calibration

The miniDOAS zero is determined by comparison to a DOAS reference instrument. Any offset in the reference instrument will show up as a similar offset in the reported miniDOAS values. The zero of the reference instrument is determined by study of a long time series, looking for periods of lowest values and assuming these occur at constant zero ammonia levels.

If this assumption is incorrect, it results in the reference instrument underestimating the real concentrations. This would therefore lead to a negative bias in the concentrations reported by the miniDOAS, never to a positive one. There is no evidence for this in the dataset.

### 4.4 Intercomparison at shorter timescales

On a timescale of minutes or even hours, the instruments do not compare well. This is caused by a distinct difference in

temporal resolution: the typical integration time of the miniDOAS is 1 minute, its minute-measurements are delay-free and mutually independent. The AMOR has a much larger response time, despite its claimed temporal resolution of 3 minutes. Its response to abrupt changes shows a delay (order of 30-60 minutes) and a spread-out and flattening of short peaks. In general, the integral over time of the AMOR-observed ammonia seems to remain conserved, as is reflected by the good comparison of longer timescale averages discussed above. This means that (virtually) no ammonia is lost in the AMOR, but it will be

recorded at a different moment in time than its actual appearance at the AMOR inlet.



On timescales of hours, e.g. when looking at daily cycles, we consider the miniDOAS concentrations to be more representative for the actual ambient ammonia concentrations than the AMOR measurements.

## 5 Conclusions

The Dutch National Air Quality Monitoring Network has been monitoring ambient ammonia concentrations since 1992, using automated AMOR instruments. Over a period of 22 years, an hourly dataset was obtained at eight stations in the Netherlands. In 2014 the number of stations was reduced to six. On 1 January 2016, six miniDOAS instruments have replaced the AMOR instruments. The DOAS technique is an open-path remote sensing technique that does not require bringing ammonia inside an instrument. This technique avoids all adverse effects typical for most commercial ammonia measurements: adsorption to tubing, filters and instrument interior, and interference from aerosols generating ammonia. In addition, a substantial reduction of operating costs is obtained.

Prior to the transition, both instruments ran in parallel at six stations for a period of 16 months. The comparison during this period shows that both instruments have a similar uptime, obtaining 80 to 90% of the possible hourly values. This is adequate for network operations.

The introduction of the miniDOAS in the Dutch Air Quality Monitoring Network results in a substantial reduction of the instrument response time and thus in a gain in temporal resolution. Consequently, miniDOAS minute values and hourly values will be more representative for ambient ammonia concentrations. The resulting dataset will be better suited for the study of daily cycles and processes than the dataset based on AMOR data. Compass analysis, i.e. sorting concentration data by episodes of a single wind direction to investigate in which direction ammonia sources are located, also becomes possible with the high temporal resolution of the miniDOAS.

Daily-averaged and especially monthly-averaged values of both instruments compare well. The miniDOAS dataset shows a small positive offset of $1.0 \pm 0.6$ µg m$^{-3}$ to the AMOR dataset. The origin of this offset is presently unknown. As a potential cause for this offset, we cannot rule out possible losses in the AMOR inlet, as this part of the instrument is not included in the calibration process. Other possibilities are: the height difference between AMOR inlet and DOAS path in combination with a deposition gradient, and the possible influence of the monitoring station housing on ammonia concentrations at the AMOR inlet.

## 6 Outlook

Over recent years, we received several requests to make a miniDOAS available to other parties. So far, we worked together with Agroscope in Switzerland (Sintermann et al., 2016) and the Flanders Environment Agency (VMM) in Belgium. As a result, some 10 miniDOAS systems are currently operational in these countries. We are now exploring the possibilities to make the miniDOAS instrument available as a commercial instrument, through collaboration with one or more partners.





To further improve the calibration, especially the zero measurements (Sect. 2.3.4), we intend to construct a laboratory facility with zero concentration over the full path length of the instrument.

We anticipate being able to measure ammonia deposition with miniDOAS soon. Using the gradient technique, we aim for hourly deposition measurements. This development will be subject of a forthcoming paper.

**7    Data availability**

The full dataset of hourly data, daily averaged data and monthly averaged data is provided in a supplement to this paper.

**Appendix A: monitoring network overview**

**Measurement station overview**

Figures A1-A6 show for each operational ammonia monitoring station an overview map and a topographic map of the
immediate surroundings.

Interactive 360° views of the monitoring stations may be found on the following website:
https://www.onsite360.nl/projecten/rivm2015/startpagina/

**Implementation of the miniDOAS in the monitoring network**

General remarks: at all sites, the optical path is at about 2.20 m above the ground. The ground is level at every site, so the
path stays at this height over its entire length. Path lengths are given as 2 x the distance between the miniDOAS and the retroreflector. Path directions are given in degrees east of north, i.e. 270° is due west. Fro each station, its ID that is used in national and international databases is given, its place name and street name.

Station: NL10131, Vredepeel, Vredeweg.
Location: latitude 51.540520° N, longitude 5.853070° E
Path length: 2 x 25.0 m.
Path direction: 344°.
Remarks: the monitoring station is on the grounds of an experimental farm. The retroreflector is mounted on the wall of one of the farm buildings.

Station: NL10444, De Zilk, Vogelaarsdreef.
Location: latitude 52.296556° N, longitude 4.510817° E
Path length: 2 x 25.0 m.
Path direction: 217°.

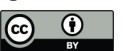



Remarks: none.

Station: NL10538, Wieringerwerf, Medemblikkerweg.

Location: latitude 52.803657° N, longitude 5.050509° E

Path length: 2 x 22.6 m (until November 2014), 2 x 18.4 m (since November 2014).

Path direction: 178° (until November 2014), 95° (since November 2014).

Remarks: In November 2014, this station was re-sited, it was moved 110 m to the west. Also, the optical path was turned 90°. As a consequence, in March and October, the instrument looks straight into the rising sun. To avoid damage to the sensitive optics, the instrument is switched off and its optics are covered during those months. Installation of an automatic

shutter that only covers the instrument when the sun is actually in the field of view is envisaged.

Station: NL10633, Zegveld, Oude Meije.

Location: latitude 52.137950° N, longitude 4.838190° E

Path length: 2 x 15.4 m.

Path direction: 171°.

Remarks: Because of a restricted site, the optical path is shorter than on other locations.

Station: NL10738, Wekerom, Riemterdijk.

Location: latitude 52.111621° N, longitude 5.708419° E

Path length: 2 x 17.6 m.

Path direction: 303°.

Remarks: The optical path passes through hoisting machinery. This has no influence on the measured concentration, as the air can move freely through it.

Station: NL10929, Valthermond, Noorderdiep.

Location: latitude 52.875725° N, longitude 6.932432° E

Path length: 2 x 14.7 m.

Path direction: 336°.

Remarks: Because of a restricted site, the optical path is shorter than on other locations.

**Appendix B: AMOR data offset**

As indicated in the main text, the AMOR data were corrected for a small offset. This offset is stable in time but varies between stations. It is produced in the digital to analogue converter (DAC) that transforms the digital AMOR signal to an



analogue signal which in turn is digitised by the data acquisition system of the station. Table 8 shows the resulting offsets per station. AMOR data used in this study have been corrected for these offsets, in contrast to the original AMOR dataset that is now present in national and international databases. A correction of the data in the official database is planned and will be documented in a separate publication.

**Author contribution**

Lou Gast and Marty Haaima installed and operated the miniDOAS instruments. Hans Verboom and Guus Stefess coordinated the introduction of the miniDOAS in the monitoring network. Marty Haaima and Stijn Berkhout processed the data. All authors were involved in the data analysis, with Theo Hafkenscheid and Ronald Hoogerbrugge performing the statistical part of the analysis. The manuscript was prepared by Stijn Berkhout, Daan Swart and Hester Volten, with

contributions from the other authors.

**Competing interests**

The authors declare that they have no conflict of interest.

**Acknowledgements**

We thank Jan Venema (RIVM) for his help in determining the AMOR data offset. We thank Addo van Pul (RIVM), Enrico

Dammers (RIVM and Free University Amsterdam) and Margreet van Zanten (RIVM) for their comments on the manuscript of this paper.

This work was carried out in the scope of the Dutch National Air Quality Monitoring Network. It was financed by the Dutch Ministry of Infrastructure and the Environment.

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

**Table 1. Overview of a number of active DOAS systems for measuring ammonia between 200-230 nm, arranged in chronological order. See the text for a discussion. RIVM: the modified commercial instrument discussed above, miniDOAS: the smaller**
10   **instrument developed later.**

| Instrument | Edner | Mount | RIVM | miniDOAS | Sintermann |
|---|---|---|---|---|---|
| detection limit ($\mu$g m$^{-3}$) | 1 | 1.3 | 0.15 | 0.25 | 0.2 |
| integration time (min) | 15 | 5 | 5 | 1 | 1 |
| set-up | bistatic | monostatic | monostatic | monostatic | monostatic |
| path length (m) | 265-350 | ~2-150 | 100 | 42 | 75 |
| light source | 150W Xe | 150W Xe | 150W Xe | 30W Xe | D$_2$ |
| detector type | scanning slit monochromator | cooled diode array | cooled CCD | CCD | cooled CCD |
| interference filter | yes | no | yes | yes | no |
| reference | Edner et al. (1990) | Mount et al. (2002) | Volten et al. (2012a) | Volten et al. (2012a) | Sintermann et al. (2016) |

**Table 2. Gas mixtures used for reference spectra. PRM: Primary Reference Material. The equivalent concentration in the atmosphere is given for a 2 x 21 m optical path, at 0° C and 100 kPa.**

| Gas | Mixing ratio in gas cell (ppm in N$_2$) | Equivalent concentration in atmosphere ($\mu$g m$^{-3}$) |
|---|---|---|
| NH$_3$ (PRM) | 300 ± 2% | 401.7 ± 2% |
| NH$_3$ | 375 ± 2% | 502.2 ± 2% |
| SO$_2$ | 30 ± 2% | 151.1 ± 2% |
| NO | 600 ± 2% | 1416 ± 2% |

15   **Table 3. Specifications for the miniDOAS, for 1-minute values.**

| | |
|---|---|
| Detection limit ($\mu$g m$^{-3}$) | 0.25 |
| Precision ($\mu$g m$^{-3}$) | 0.25 |
| Time resolution (min) | 1 |





**Table 4. Systematic error sources in the miniDOAS measurements.**

| Systematic error source | Magnitude |
|---|---|
| path length determination | ± 0.25% |
| calibration gas concentration | ± 2% |
| reference DOAS concentration | 0.45 µg m$^{-3}$ |

**Table 5. Height difference assessment with passive samplers. For each month in 2014, the bullets (•) indicate that measurements were taken at the respective stations.**

| Station name | Jan | Feb | Mar | Apr | May | Jun | Jul | Aug | Sep | Oct | Nov | Dec |
|---|---|---|---|---|---|---|---|---|---|---|---|---|
| Vredepeel | • | • | • | • | • | • | | | | | | |
| De Zilk | • | • | • | • | • | • | • | • | • | • | • | • |
| Wieringerwerf | | | | | | | • | • | • | • | • | • |
| Zegveld | | | • | • | • | • | • | • | • | • | • | • |
| Eibergen | • | • | • | • | • | • | • | • | • | • | • | • |
| Valthermond | | | | | | | • | • | • | • | • | • |

**Table 6. Average hour values for the NH$_3$ concentrations over the period from 1-9-2014 to 31-12-2015, based on pairs of hourly values. Standard deviations (σ) of the averages are given as well. N: the number of data pairs used to calculate the averages. The overall average bias is 1.0 ± 0.6 µg m$^{-3}$.**

| Station name | AMOR average concentration (µg m$^{-3}$) | miniDOAS average concentration (µg m$^{-3}$) | N |
|---|---|---|---|
| Vredepeel | 16.8 (σ = 14.3) | 18.4 (σ = 20.6) | 6429 |
| De Zilk | 2.1 (σ = 2.5) | 3.0 (σ = 2.9) | 8130 |
| Wieringerwerf | 4.7 (σ = 4.7) | 5.6 (σ = 6.2) | 6824 |
| Zegveld | 8.6 (σ = 7.3) | 8.6 (σ = 7.7) | 8554 |
| Wekerom | 12.9 (σ = 11.5) | 14.5 (σ = 13.1) | 8156 |
| Valthermond | 4.7 (σ = 3.8) | 5.9 (σ = 4.2) | 8711 |

**Table 7. Annual averaged differences in ammonia concentrations measured with passive samplers at different measurement heights averaged over all sites.**

| Difference of passive sampler averages | Difference (µg m$^{-3}$) | Remarks |
|---|---|---|
| AMOR inlet (3.5 m) – Mast 2.2 m | 0.0 ± 0.5 (n = 47) | Possible offset between AMOR and DOAS due to height difference |





| Difference of passive sampler averages | Difference (µg m$^{-3}$) | Remarks |
|---|---|---|
| Mast 3.5 m – Mast 2.2 m | 0.5 ± 0.3 (n = 50) | Possible gradient due to deposition |
| Mast 3.5 m – AMOR inlet (3.5 m) | 0.5 ± 0.6 (n = 47) | Expected to be zero, unless e.g. influence of station housing on AMOR measurement |

**Table 8. Offset due to digital-to-analogue conversion, per monitoring station. This is defined as follows: (reconstructed digital signal) = (signal after digital-to-analogue conversion) + offset.**

| Station name | offset (µg m$^{-3}$) |
|---|---|
| Vredepeel | 0.77 ± 0.19 |
| Huijbergen | 0.13 ± 0.02 |
| De Zilk | 0.02 ± 0.02 |
| Wieringerwerf | 0.22 ± 0.03 |
| Zegveld | 0.92 ± 0.11 |
| Eibergen | 0.97 ± 0.10 |
| Wekerom | 0.72 ± 0.09 |
| Valthermond | 0.14 ± 0.02 |

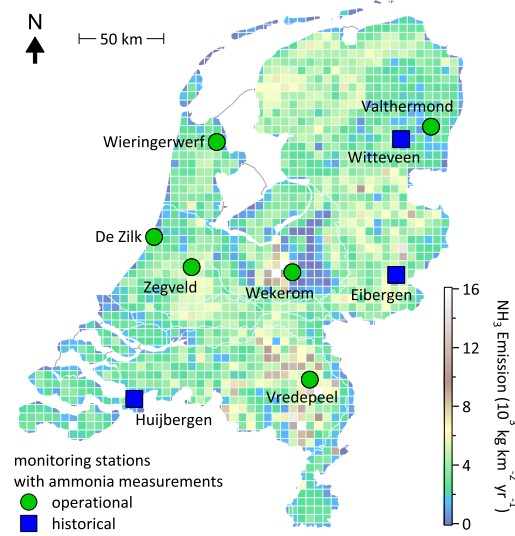

**Figure 1: Map of air quality monitoring locations with ammonia measurements in the Netherlands, shown as green circles. Historical locations are shown as blue squares. Shown in the background is an emission map, with emissions in 2013 derived from**





the Dutch emission inventory, on a 5x5 km grid. Witteveen was decommissioned in 2000, its observations were transferred to Valthermond. Ammonia measurements at Eibergen and Huijbergen were terminated in 2014.

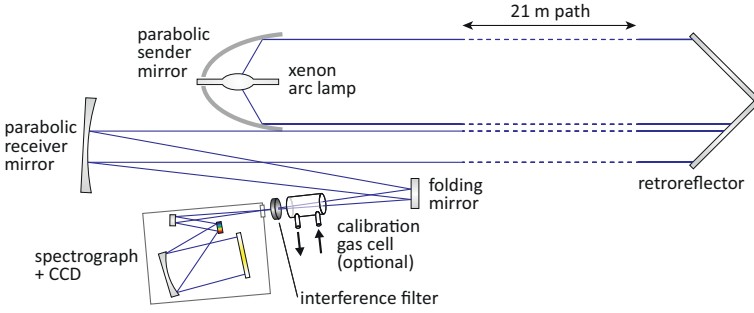

5   **Figure 2: Schematic representation of the miniDOAS, with an optional calibration gas cell in the optical path.**

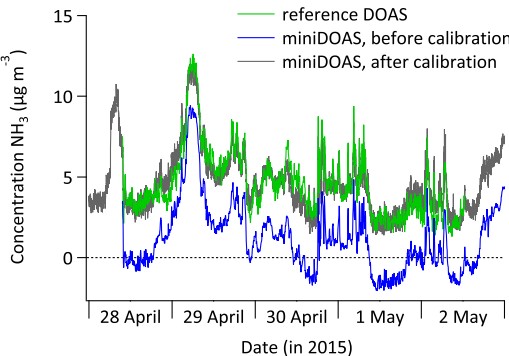

**Figure 3: Example of miniDOAS and reference DOAS measurement. Reference DOAS: the concentrations as measured by the reference DOAS instrument. miniDOAS, before calibration: the concentrations as measured by the miniDOAS, without correction**
10   **for a proper reference spectrum $I_{background}(\lambda)$. miniDOAS, calibrated data: the concentrations as measured by the miniDOAS, now with correction for a proper reference spectrum $I_{background}(\lambda)$, made from spectra from this period and calibrated with the concentration measured by the reference DOAS.**





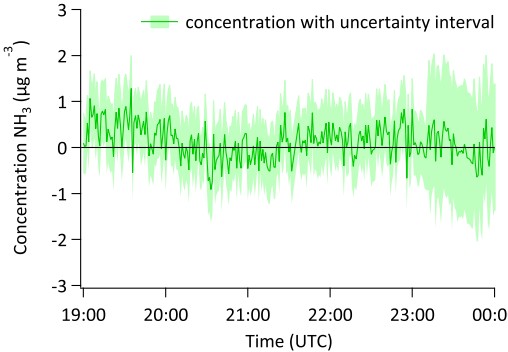

**Figure 4: Example of an episode of very low ambient ammonia concentrations, as measured with the miniDOAS at Wieringerwerf on 16 August 2015. Minute values of the measured ammonia concentrations in µg m⁻³ are indicated by a green line, the light green area around this line indicates the 1-sigma uncertainty interval.**

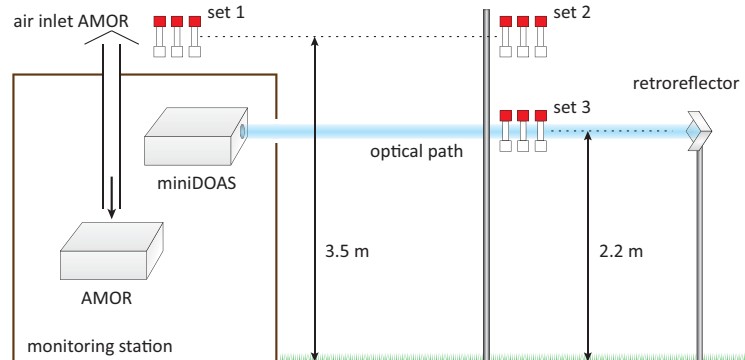

**Figure 5: Set-up of three sets of three Gradko passive samplers for NH₃ at each of the six miniDOAS stations, aiming to quantify a possible systematic difference caused by measurement height. Set 1: attached to the AMOR air inlet, 3.5 m height. Set 2: in a mast halfway the miniDOAS optical path (of typically 21 m length), at 3.5 m height (i.e. the AMOR air inlet height). Set 3: in the same mast, at 2.2 m height (i.e. the miniDOAS optical path height).**



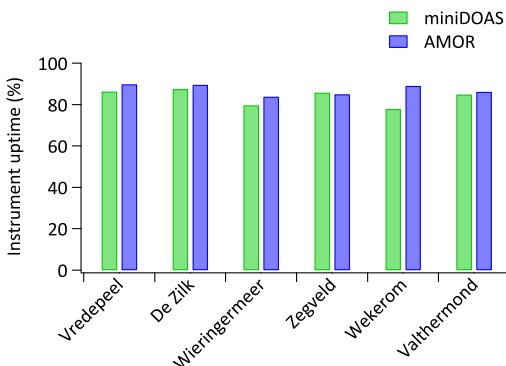

**Figure 6: Uptimes of the AMOR and miniDOAS systems over the period from 1-9-2014 to 1-9-2015, excluding periods where the miniDOAS systems were in the laboratory for instrument characterisation, when station housings were renewed, et cetera.**

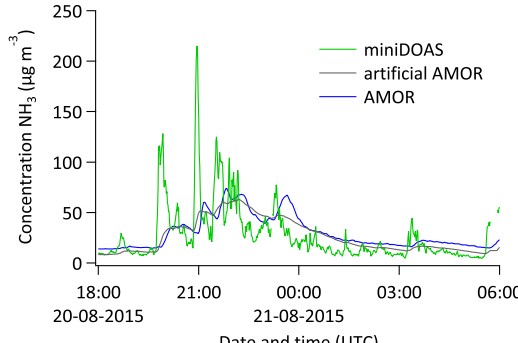

**Figure 7: Vredepeel, miniDOAS minute data converted to artificial AMOR data by delaying and smoothing. Here we used an e-folding time $\tau_{1/e}$ of an hour. The delayed and smoothed miniDOAS data becomes remarkably similar to the AMOR data. Note that the surfaces under the various curves are more or less similar, indicating that both instruments on average detect similar amounts of ammonia.**





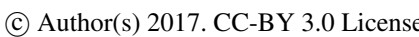

**Figure 8: Hourly ammonia concentrations measured at the monitoring station in Vredepeel from 15 to 22 March 2014. Inset: concentrations measured on 17 March between 0:00 and 18:00 UTC.**

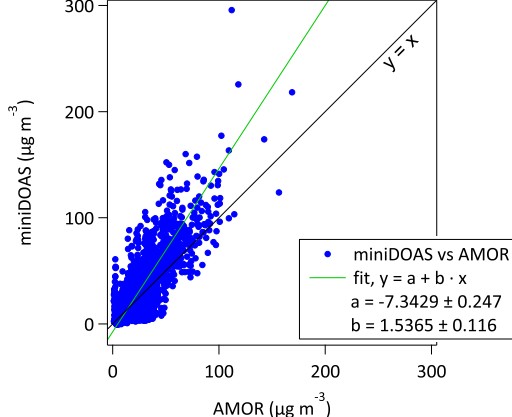

**Figure 9: Scatter plot (6429 data points) of hourly ammonia concentrations measured at the monitoring station in Vredepeel for 1 September 2014 up to and including 31 December 2015. The fit shown is a least orthogonal distance fit.**





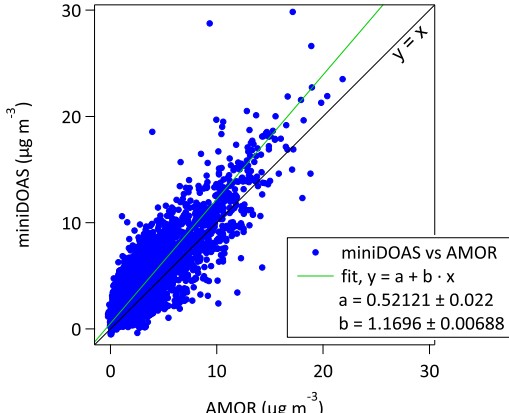

**Figure 10: Scatter plot (8130 data points) of hourly ammonia concentrations measured at the monitoring station in De Zilk for 1 September 2014 up to and including 31 December 2015. The fit shown is a least orthogonal distance fit.**

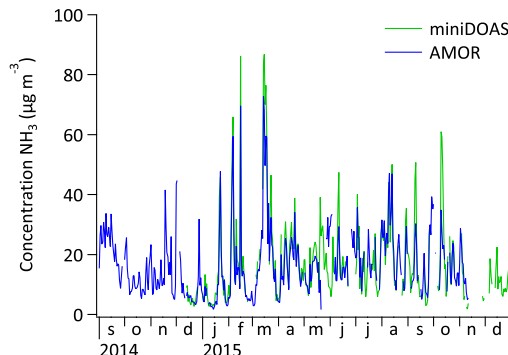

**Figure 11: Daily ammonia concentrations measured at the monitoring station in Vredepeel for 1 September 2014 up to and including 31 December 2015.**




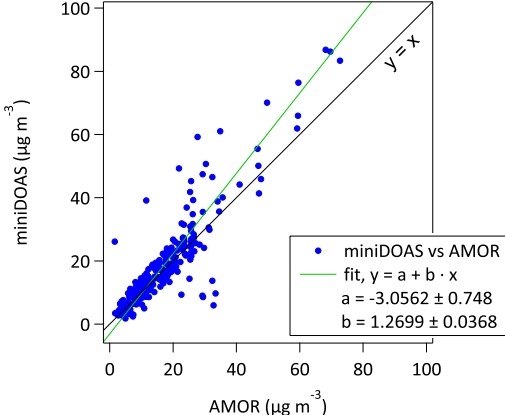

**Figure 12: Scatter plot (259 data points) of daily ammonia concentrations measured at the monitoring station in Vredepeel for 1 September 2014 up to and including 31 December 2015. The fit shown is a least orthogonal distance fit.**

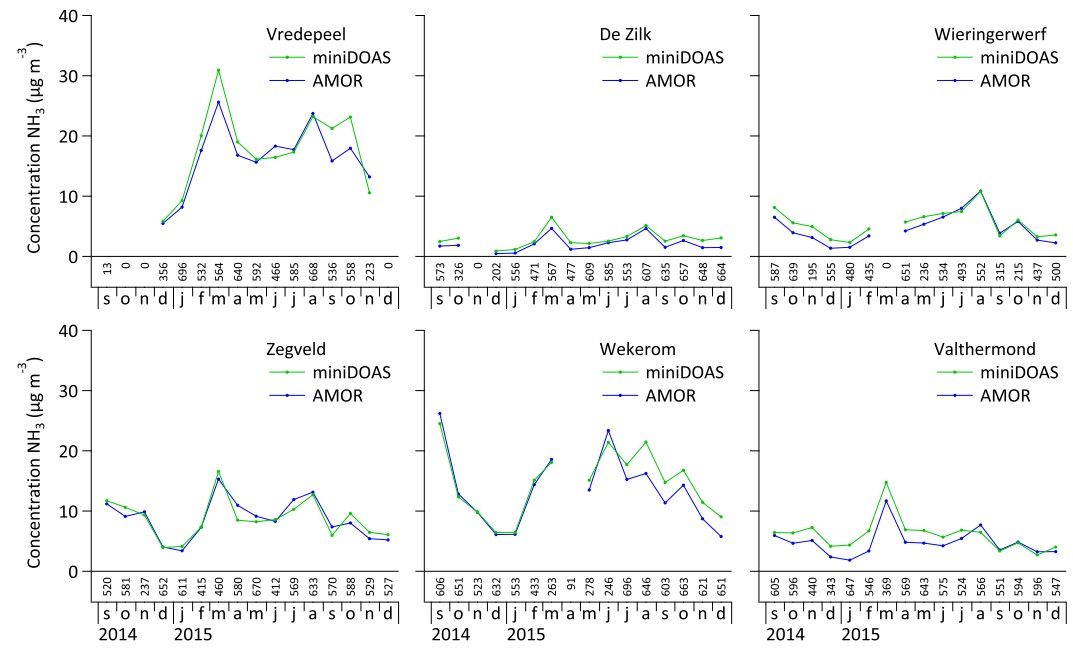

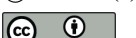


**Figure 13: Monthly averaged NH₃ concentrations measured with the AMOR and miniDOAS for the six ammonia measurement stations of the LML. The numbers above the x-axes indicate the number of hours in that month with a concentration value available for both instruments. A monthly average was included if there were at least 100 hourly values.**

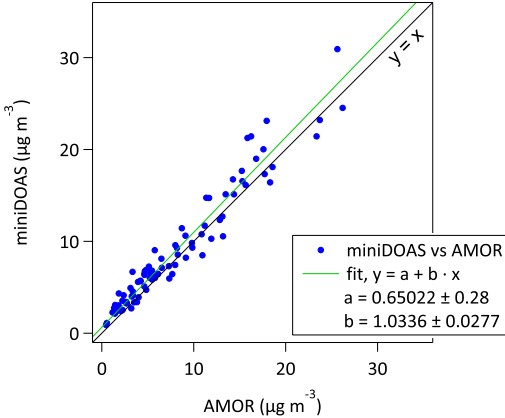

**Figure 14: Scatter plot (89 data points, R²=0.94) of monthly ammonia concentrations measured at all stations combined for 1 September 2014 up to and including 31 December 2015. The fit shown is a least orthogonal distance fit.**

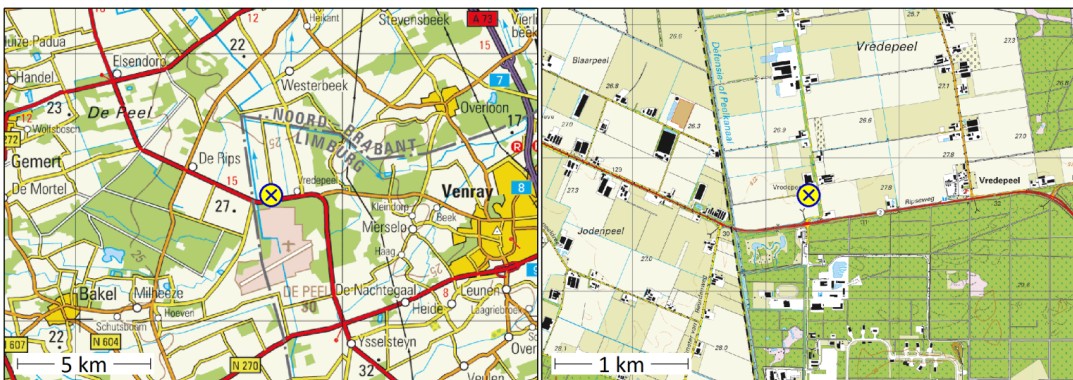

10     **Figure A1: Maps of monitoring station 131, Vredepeel. The location of the station is indicated with ⊗. Station environment: Arable land, livestock housing, forest.**




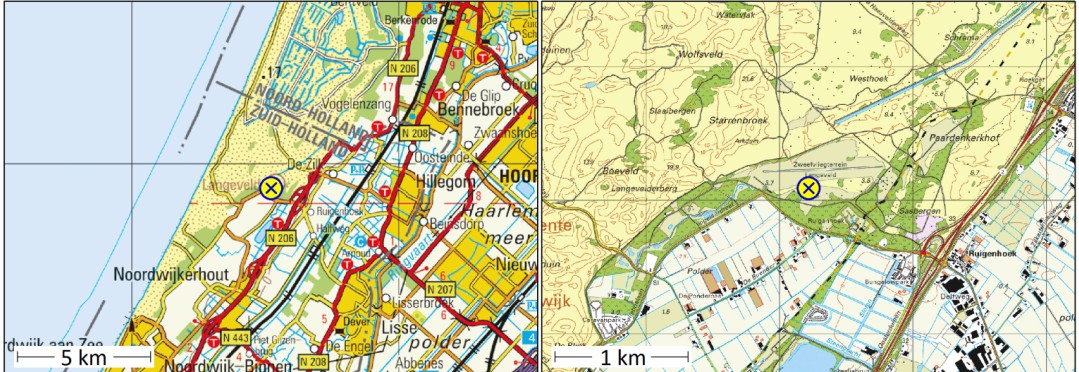

**Figure A2: Maps of monitoring station 444, De Zilk. The location of the station is indicated with ⊗. Station environment: Dunes, coast.**

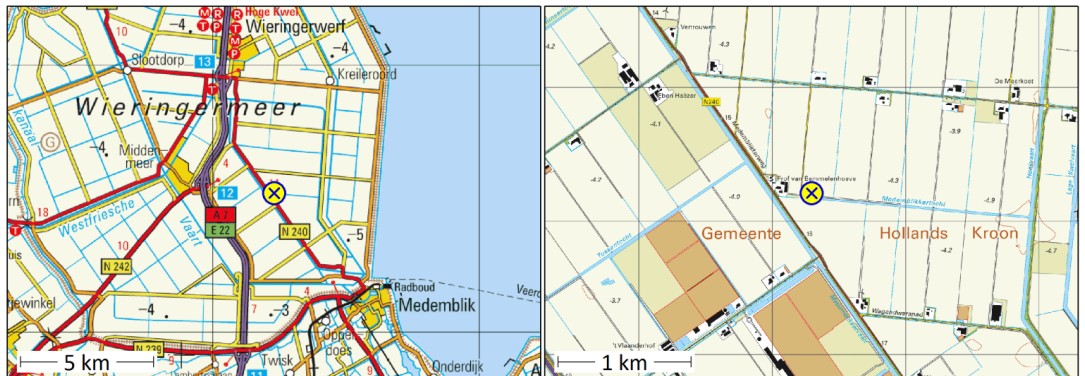

5    **Figure A3: Maps of monitoring station 538, Wieringerwerf. The location of the station is indicated with ⊗. Station environment: Arable land.**



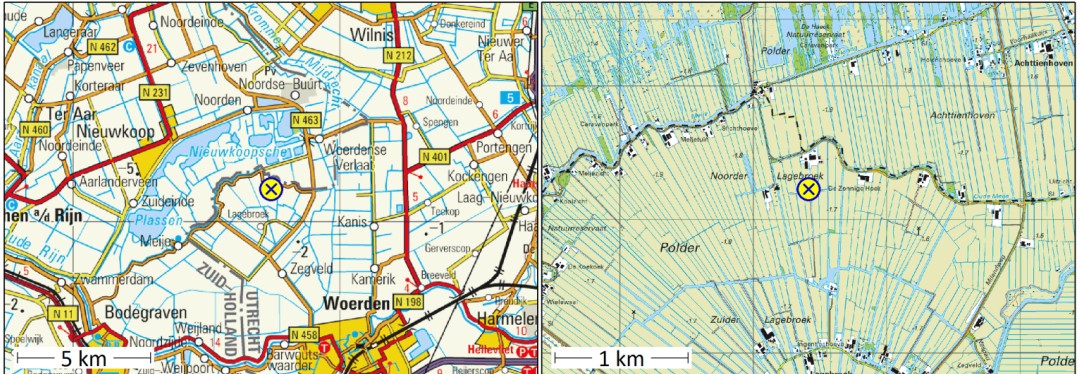

**Figure A4: Maps of monitoring station 633, Zegveld. The location of the station is indicated with ⊗. Station environment: Pastures.**

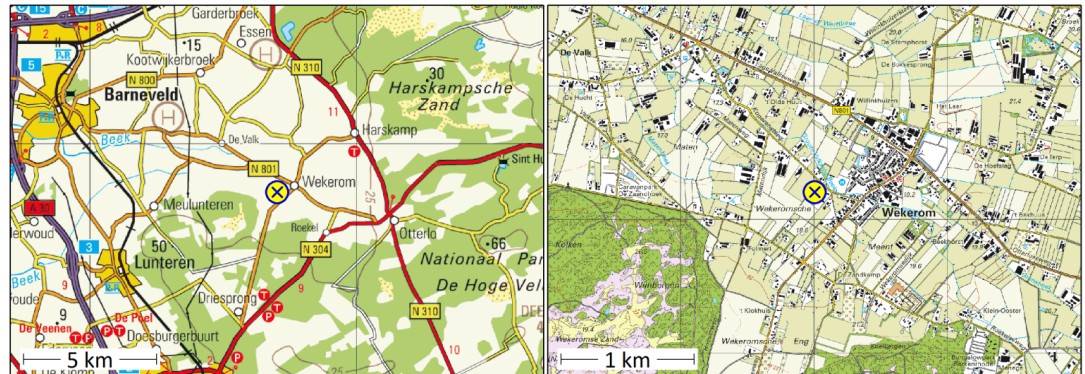

5   **Figure A5: Maps of monitoring station 738, Wekerom. The location of the station is indicated with ⊗. Station environment: Arable land, pastures, forest.**




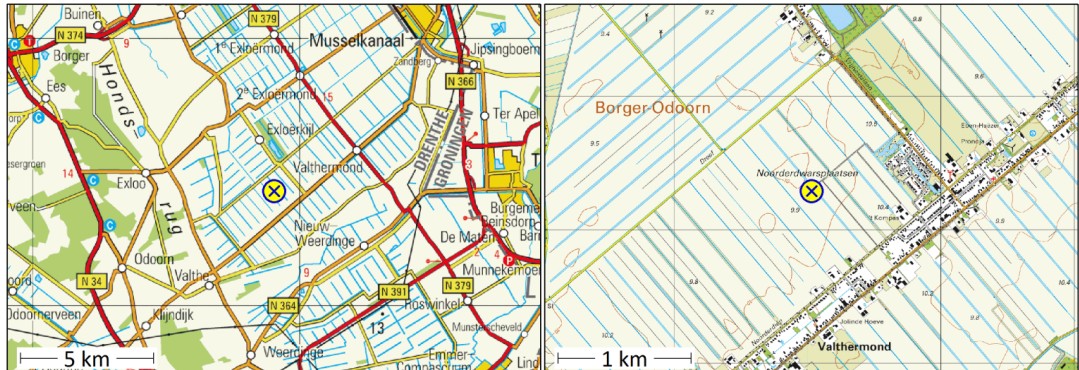

**Figure A6: Maps of monitoring station 929, Valthermond. The location of the station is indicated with ⊗. Station environment: Arable land.**