# Peer review of "Replacing the AMOR by the miniDOAS in the ammonia monitoring network in the Netherlands"

_Atmospheric Measurement Techniques, 2016_

## Referee Comment (RC1) · A. Neftel (Referee) · 21 Mar 2017

As I have a long lasting collaboration with the authors regarding the further development of MiniDOAS system to measure ambient NH3 concentrations this is not an "official" review.

I congratulate the Dutch authorities that they had the courage to replace the well-established AMOR systems ( a wet-chemistry based point monitor) with the newly developed MiniDOAS systems (even though this seems more driven by financial then scientific reasons). To avoid inlet systems and wet chemistry analytic systems is a major advantage for ambient NH3 measurements. The older systems all show varying smoothing behavior that hardly can be quantified and corrected. The new DOAS systems do provide real-time 1 minute values if there is no fog (or tractors or elephants) in

the path. The replacement of the AMOR system occurred over a 15-month period with overlapping measurements on 6 stations. This allows an excellent comparison and an evaluation of the continuity that is crucial for a monitoring network. I see still two critical issues in the NH3 concentration evaluation of MiniDOAS measurements: a) Accuracy of the sensitivity (i.e. the span of the calibration) b) Accuracy of the offset (related to the number density of NH3 molecules on the reference path)

To a) To determine the NH3 calibration spectrum the authors used a short flow cell cuvette in the path flushed with a 300ppm gas concentration from a certified standard. As NH3 is a very sticky compound it is not a very safe assumption that equilibrium is reached. I propose to analyse the time course of the measured DOAS concentration from the beginning and to monitor the concentration after the cell with an independent approach. Our own investigations show that (for our setup) equilibrium is reached only after at least one hour.

To b) The first step in the evaluation of the recorded spectra is the division by a reference spectra, both corrected by the dark current. In an ideal world, the reference spectra would be taken on the same conditions as the standard measurements but without any absorbing gas. The evaluated concentration then corresponds to the difference between the mean concentration on the path and the concentration of the reference spectra. Consequently, negative concentrations are physically possible and should be reported as such. It would be worthwhile to establish a clean air facility that allows to measure over normally used path length with concentrations well below the detection limit (e.g. 0.05 ppb for NH3). Practically this is not yet possible and ways around this described by the authors and as by Sintermann et al. (AMT 2016) must be taken. Doing so accuracy better than 0.5 ppb can hardly be reached. Monitoring networks that aim at real-time value accessible over the web, will have difficulties with this issue and demand always positive values. This easily puts a scientifically not justifiable boundary conditions on DOAS measurements.

The 15 month intercomparison on six stations is a very interesting data set that allows a

deep insight into the performance of NH3 monitoring systems that cannot be gained by the different intercomparison exercises that has been published up to now (see e.g. von Bobrutzki et al., Atmos.Meas. Tech., 3, 91–112, 2010,; Milford et al., Biogeosciences, 6, 819-834, 2009). The overall correspondence between the two systems based on the ensemble of monthly means is remarkably well. But going into details, doubts on this well behaving come up. In principle going to shorter integration intervals should not change the overall picture, but should increase the scatter. The used regression (least orthogonal distance fit) could be misleading as the high DOAS concentrations that are smoothed by the AMOR systems are kind of outliers. I guess that using a robust approach (e.g. "pbreg" or "deming" in the R-world) would give comparable slopes and offset for all three time resolutions.

In case I analyze the six stations separately I see persistent offset and slopes clearly different from 0 resp. 1 over several months, pointing of systematic drifts either in the AMOR and/or MiniDOAS systems. I do propose to analyse separately the time periods between service intervals of the AMOR systems as these actions constitutes a kind of reset of these instruments that might cause a change in the response function. The same should be done, in case a DOAS component is changed (lamp, reflector, etc). As the DOAS systems are new we must learn about their long-term behavior and be open of surprises that might affect the performance. The presented dataset is an excellent playground. The DOAS technique also has the great advantage that concentration evaluation can be repeated in case new knowledge is available.

Albrecht Neftel, Neftel Research Expertise, CH 3033 Wohlen b. Bern, Switzerland

---

## Referee Comment (RC2) · Anonymous Referee #3 · 27 Jun 2017

The manuscript by Berkhout et al. titled "Replacing the AMOR by the miniDOAS in the ammonia monitoring network in the Netherlands" describes a network of the LP-DOAS instruments for measuring ammonia installed as a part of the Ditch National Air Quality Monitoring Network. The DOAS instruments were replacing an outdated AMOR instruments, which quality ammonia in-situ using wet chemistry. The manuscript also analyzes a dataset collected during a 16-months-long intercomparison measurements between DOAS and AMOR instruments at six sites throughout the country.

Overall, I was pleased to learn about the Dutch air authorities using optical remote sensing (ORS) technology for air monitoring. ORS methods, including DOAS, offer a number of advantages over "traditional" measurement techniques, however, they have been largely underutilized by governmental agencies. This manuscript provides an ex-

ample of how DOAS method can be successfully used for contentious, long-term, near real-time measurements by a regulatory agency; and hence, this article is interesting for publication. I also envision that this paper will be of interest for scientists who rely on the Dutch ammonia network data for their research. However, in my opinion, this paper has a number of deficiencies that have to be addressed, before paper can be published.

Sections below provide a list of specific comments for authors to address. Additionally, I believe that this manuscript can greatly benefit from English language editing. I encourage the authors to take advantage of the English editing services offered by AMT, or have the manuscript edited by a native English speaker.

Finally, I think that paper can be significantly strengthened by expanding the analysis of the collected data, data interpretation, and discussion of the results.

Major comments

Section 2.3.2 – Add a figure with the example of the DOAS spectral fit. List all parameters included in spectral evaluation.

One of the major strengths of the DOAS technique is that calibration is not necessary and concentrations can be determined from first principles; however, authors significantly diminished such advantage by devising a cumbersome method for measuring a reference spectra. Authors should explain why they chose to use reference calibration rather than spectral fitting using literature absorption cross-section convoluted with the instrument function.

Authors should provide a more detailed explanation for the reference DOAS instrument, as well as highlight how it differs from the miniDOAS's used in the network.

I am also surprised that authors did not encounter issues with using an uncooled CCD array. On page 3 authors briefly mention that use of an uncooled array leads to certain sacrifice of the instrument's performance (page 3, lines 11-12), but later on the same

page (page 3, line 28) authors state that choice of the CCD instrument performance.

It is clear that 1 minute data reporting for AMOR instruments is meaningless, I therefore recommend to remove 1min comparison between the DOAS and AMOR datasets as it bears no statistical significance. Instead, it should be highlighted that transition to the network of DOAS instruments will result in a dataset with much higher temporal resolution.

Authors should expand the description and discussion of data intercomparison with passive samplers. Table 5 should be replaced with a table of monthly averaged ammonia concentrations measured by passive samplers. Detection limits for passive samplers shall be stated. Analysis/intercomparison of passive samplers and monthly averaged DOAS and AMOR ammonia concentrations also should be presented.

Authors should provide R2 values for correlation plots in Figures 9 and 12.

I imagine that such a long intercomparison of co-located AMOR and DOAS instruments was partially conducted in order to aid future interpretation of long-term ammonia concentration trends. For this purpose, it would be also interesting to see a more in-depth analysis of the collected data. For example – the analysis of seasonal and geographical trends and/or differences in AMOR and DOAS datasets. Figure 13 shows monthly averages obtained by both instruments at all stations throughout the country. It is obvious that datasets for some stations agree better than for others. I found it disappointing that analysis was only limited to a correlation plot.

Authors should add s description of the expected maintenance schedule and expected lifetime for DOAS instruments and their major components for the DOAS instruments.

Minor comments

Replace "Uptime" with "Data capture rate"

Replace "life-expired" with outdated in reference to old instrumentation; and with "burned-out" in reference to burned-out LP-DOAS lamp

Page 2, line 16 (and through the rest of the document) - abbreviations shall be presented in parenthesis when first used in the text, and not another way around. E.g. – Differential Optical Absorption Spectroscopy (DOAS).

Page 4 – combine sections 2.2.1 through 2.2.4 into single section 2.2.

Page 6, Equation 6 - should be 2l in denominator.

Page 10, line 28 – replace "overview of the situation" with "measurements setup".

Pages 13-15 – combine section 4.1 and 4.2, and 4.4.

Pages 14-15 – eliminate sub-section in 4.3 and make a more fluid narrative (e.g. section 4.3.1 can be reduced to a single sentence or illuminated completely as it was already discussed in section 3.2).

Please provide explanation for describing the DOAS instruments deployed in the network as miniDOAS. What is the main feature that differentiates the miniDOAS from a "traditional" LP-DOAS instrument? Table 1 can be augmented to include physical dimensions of the different instruments.

---

## Author Comment (AC1) · 25 Jul 2017

(comment from referee)

As I have a long lasting collaboration with the authors regarding the further development of MiniDOAS system to measure ambient NH3 concentrations this is not an "official" review. I congratulate the Dutch authorities that they had the courage to replace the well- established AMOR systems ( a wet-chemistry based point monitor) with the newly developed MiniDOAS systems (even though this seems more driven by financial then scientific reasons). To avoid inlet systems and wet chemistry analytic systems is a major advantage for ambient NH3 measurements. The older systems all show varying smoothing behavior that hardly can be quantified and corrected. The new DOAS sys-

tems do provide real-time 1 minute values if there is no fog (or tractors or elephants) in the path. The replacement of the AMOR system occurred over a 15-month period with overlapping measurements on 6 stations. This allows an excellent comparison and an evaluation of the continuity that is crucial for a monitoring network. I see still two critical issues in the NH3 concentration evaluation of MiniDOAS measurements: a) Accuracy of the sensitivity (i.e. the span of the calibration) b) Accuracy of the offset (related to the number density of NH3 molecules on the reference path) To a) To determine the NH3 calibration spectrum the authors used a short flow cell cuvette in the path flushed with a 300ppm gas concentration from a certified standard. As NH3 is a very sticky compound it is not a very safe assumption that equilibrium is reached. I propose to analyse the time course of the measured DOAS concentration from the beginning and to monitor the concentration after the cell with an independent approach. Our own investigations show that (for our setup) equilibrium is reached only after at least one hour.

(author's response)

We allow the system to reach a steady state for 30 minutes, we then take data during 1 hour. We find that after 11 minutes the concentration is within 2% of the average concentration measured during the hour of actual data-taking. After 25 minutes, it is within 0.5%. Note that the concentration in the gas bottle is known with 2% accuracy, according to the gas manufacturer. The difference between our set-up and yours may arise from a larger flow, shorter tubing, et cetera. We insert the following sentence in the manuscript (page 7, line 8):

(author's change in manuscript)

We find that, after 25 minutes, the concentration in the cell is within 0.5% of the final concentration. We therefore allow the system 30 minutes to reach this steady state, then we collect spectra during 1 hour.

(comment from referee)

To b) The first step in the evaluation of the recorded spectra is the division by a reference spectra, both corrected by the dark current. In an ideal world, the reference spectra would be taken on the same conditions as the standard measurements but without any absorbing gas. The evaluated concentration then corresponds to the difference between the mean concentration on the path and the concentration of the reference spectra. Consequently, negative concentrations are physically possible and should be reported as such.

(author's response)

We do find negative concentrations and we do report them. In the entire dataset of 53908 hourly values (see also the supplement), 163 or 0.30% were negative. Averaging of the data leads to fewer negative values, as can be expected; we found no negative values in the daily or monthly averages.

(comment from referee)

It would be worthwhile to establish a clean air facility that allows to measure over normally used path length with concentrations well below the detection limit (e.g. 0.05 ppb for NH3). Practically this is not yet possible and ways around this described by the authors and as by Sintermann et al. (AMT 2016) must be taken. Doing so accuracy better than 0.5 ppb can hardly be reached. Monitoring networks that aim at real-time value accessible over the web, will have difficulties with this issue and demand always positive values. This easily puts a scientifically not justifiable boundary conditions on DOAS measurements.

(author's response)

As outlined above, we do report negative concentrations.

(comment from referee)

The 15 month intercomparison on six stations is a very interesting data set that allows a deep insight into the performance of NH3 monitoring systems that cannot be gained by

the different intercomparison exercises that has been published up to now (see e.g. von Bobrutzki et al., Atmos.Meas. Tech., 3, 91–112, 2010,; Milford et al., Biogeosciences, 6, 819-834, 2009). The overall correspondence between the two systems based on the ensemble of monthly means is remarkably well. But going into details, doubts on this well behaving come up. In principle going to shorter integration intervals should not change the overall picture, but should increase the scatter. The used regression (least orthogonal distance fit) could be misleading as the high DOAS concentrations that are smoothed by the AMOR systems are kind of outliers. I guess that using a robust approach (e.g. "pbreg" or "deming" in the R-world) would give comparable slopes and offset for all three time resolutions.

(author's response)

Thank you for the suggestion. We compared the effect of a robust regression (Winsoring the differences at 2 standard deviations) to the least orthogonal distance fit. This confirms your expectation, the slopes of the regressions of the shorter timescale fits reduce (closer to 1) and the intercepts increase (closer to 0). These results are not elaborated in the paper because the comparison of hourly values is physically rather meaningless due to the memory effects observed in the AMOR.

(comment from referee)

In case I analyze the six stations separately I see persistent offset and slopes clearly different from 0 resp. 1 over several months, pointing of systematic drifts either in the AMOR and/or MiniDOAS systems. I do propose to analyse separately the time periods between service intervals of the AMOR systems as these actions constitutes a kind of reset of these instruments that might cause a change in the response function. The same should be done, in case a DOAS component is changed (lamp, reflector, etc).

(author's response)

We performed the inverse analysis: we looked whether events in the dataset correlated

with AMOR service events. We found no such correlation. The analyses you propose come with difficulties. The AMOR systems were serviced every six weeks. Because intercomparison between AMOR and DOAS on short timescales are not feasible, this six-week period is too short to do meaningful comparisons. The miniDOAS systems, on the other hand, were exchanged only once or twice during the campaign. This makes intercomparisons difficult as well, especially when an exchange took place close to the beginning or end of the campaign period. Such an analysis could be the scope of a follow-up paper, once the miniDOAS systems have been running for 5 or 10 years.

(comment from referee)

As the DOAS systems are new we must learn about their long-term behavior and be open of surprises that might affect the performance. The presented dataset is an excellent playground. The DOAS technique also has the great advantage that concentration evaluation can be repeated in case new knowledge is available.

(author's response)

We wholeheartedly agree. This concentration re-evaluation is possible because the unprocessed spectra are stored. We would like to add a line summarising your comment to the outlook section in the manuscript (page 17, line 3):

(author's change in manuscript)

It should be noted that the miniDOAS instruments store the unprocessed spectra, averaged over 1 minute intervals. This means that reanalysis of the data, taking into account the latest insights, is always possible.

---

## Author Comment (AC2) · 25 Jul 2017

(comment from referee)

The manuscript by Berkhout et al. titled "Replacing the AMOR by the miniDOAS in the ammonia monitoring network in the Netherlands" describes a network of the LPDOAS instruments for measuring ammonia installed as a part of the Ditch National Air Quality Monitoring Network. The DOAS instruments were replacing an outdated AMOR instruments, which quality ammonia in-situ using wet chemistry. The manuscript also analyzes a dataset collected during a 16-months-long intercomparison measurements between DOAS and AMOR instruments at six sites throughout the country. Overall, I was pleased to learn about the Dutch air authorities using optical remote sensing

(ORS) technology for air monitoring. ORS methods, including DOAS, offer a number of advantages over "traditional" measurement techniques, however, they have been largely underutilized by governmental agencies. This manuscript provides an ex ample of how DOAS method can be successfully used for contentious, long-term, near real-time measurements by a regulatory agency; and hence, this article is interesting for publication. I also envision that this paper will be of interest for scientists who rely on the Dutch ammonia network data for their research. However, in my opinion, this paper has a number of deficiencies that have to be addressed, before paper can be published. Sections below provide a list of specific comments for authors to address. Additionally, I believe that this manuscript can greatly benefit from English language editing. I encourage the authors to take advantage of the English editing services offered by AMT, or have the manuscript edited by a native English speaker.

(author's response)

AMT offers English language copy-editing for final revised papers. We will gladly make use of this service.

(comment from referee)

Finally, I think that paper can be significantly strengthened by expanding the analysis of the collected data, data interpretation, and discussion of the results.

(author's response)

The aim of this paper is to present a comparison between the two instruments, in order to assess any differences in performance and to understand a discontinuity in the ammonia dataset at the changeover point, if one should arise. Further analysis is certainly interesting but is outside the scope of this paper. Please note that the entire dataset is published, as a supplement to this paper, so anyone interested in doing a further analysis can do so.

(comment from referee)
Major comments Section 2.3.2 – Add a figure with the example of the DOAS spectral fit. List all parameters included in spectral evaluation.

(author's response)

We add a Figure with all the steps in the spectral fit. This Figure is included at the end of this author's response.

(comment from referee)

One of the major strengths of the DOAS technique is that calibration is not necessary and concentrations can be determined from first principles; however, authors significantly diminished such advantage by devising a cumbersome method for measuring a reference spectra.

(author's response)

We do this by solving problems that may not be present in all DOAS applications, but they are in ours:

- By using a reference spectrum measured on the instrument itself, any pixel-to-pixel variability in sensitivity is automatically taken into account. This is a source of noise if it is ignored.

- Over the spectral range we use for our analysis, the light intensity of the observed spectrum increases by a factor of 20. Dividing by a lamp reference spectrum greatly reduces the dynamic range, this makes the approximation of the differential initial intensity l'0(lambda) more robust.

- We do agree that the calibration procedure can be improved. That is why we envisage the construction of a laboratory facility with zero concentration over the full path length of the instrument (page 17, lines 1-2).

(comment from referee)

AMTD
Authors should explain why they chose to use reference calibration rather than spectral fitting using literature absorption cross-section convoluted with the instrument function.

(author's response)

Using literature absorption spectra is only one way to construct reference spectra, the method we use is equally well accepted (CEN, 2013). All authors of the literature we cite on page 7, lines 1-4, use the same method as we do. We agree that we do not list the factors that were decisive for us to select this method. We add two sentences to clarify this (page 7, line 6):

(author's change in manuscript)

We selected this method because the spectral modelling method offers many opportunities for errors to enter the calculations, if any of the required parameters are not precisely known. The gas cell method eliminates all these potential errors. In addition, using the gas cell method, the calibration becomes traceable to a standard with a known accuracy. This is an advantage when an instrument is to be used in a monitoring network that operates under a quality management system, e.g. ISO/IEC 17025. For the miniDOAS, we use this method as well, for the same reasons.

(comment from referee)

Authors should provide a more detailed explanation for the reference DOAS instrument, as well as highlight how it differs from the miniDOAS's used in the network.

(author's response)

A more detailed description of this instrument is given in Volten et al. (2012), we refer to this in the text. We add the differences with the miniDOASes to the text (page 7, line 28):

(author's change in manuscript)

Its differences from the miniDOAS systems are the following:
- The RIVM DOAS has a better spectral resolution than the miniDOAS (0.0306 nm vs. 0.067 nm).

- The RIVM DOAS uses a cooled CCD detector, the CCD of the miniDOAS is not cooled.

- The wavelength calibration of the RIVM DOAS is continuously monitored by measuring the emission line of a zinc lamp. If needed, the grating of the spectrograph is adjusted. The miniDOAS has no such option.

- The RIVM DOAS reports values at 5 minute intervals, the miniDOAS at 1 minute intervals.

(comment from referee)

I am also surprised that authors did not encounter issues with using an uncooled CCD array. On page 3 authors briefly mention that use of an uncooled array leads to certain sacrifice of the instrument's performance (page 3, lines 11-12), but later on the same page (page 3, line 28) authors state that choice of the CCD instrument performance.

(author's response)

This sentence seems to be missing a part. We guess that you find our statement that the performance is not unduly affected a bit brief, we agree with that. We added a short remark (page 3, line 28):

(author's change in manuscript)

The use of an uncooled CCD increases the noise in the spectra, but we find that this does not affect the retrieval of the concentrations unduly.

(comment from referee)

It is clear that 1 minute data reporting for AMOR instruments is meaningless, I therefore recommend to remove 1min comparison between the DOAS and AMOR datasets as

AMTD
it bears no statistical significance. Instead, it should be highlighted that transition to the network of DOAS instruments will result in a dataset with much higher temporal resolution.

(author's response)

We show the 1 minute comparison to substantiate our claim that it is meaningless, we are therefore reluctant to remove it. The higher temporal resolution of the miniDOAS dataset is already highlighted in the Conclusions section, please see page 16, lines 14 to 19.

(comment from referee)

Authors should expand the description and discussion of data intercomparison with passive samplers. Table 5 should be replaced with a table of monthly averaged ammonia concentrations measured by passive samplers. Detection limits for passive samplers shall be stated. Analysis/intercomparison of passive samplers and monthly averaged DOAS and AMOR ammonia concentrations also should be presented.

(author's response)

All details of the passive samplers are given in the paper by Lolkema et al. (2015), to which we refer. We added a sentence to the paper to make this more clear (page 10, line 28):

(author's change in manuscript)

The passive samplers were operated as they are in the Measuring Ammonia in Nature network. All information on these samplers – handling, detection limits, calibration et cetera – can be found in Lolkema et al. (2015).

(author's response)

The passive sampler measurements presented here are meant only to investigate a possible difference in concentrations at various heights. An intercomparison with the
passive samplers described in this section is not possible, because these samplers are calibrated against the monitoring network measurements (see (Lolkema et al., 2015)). A full intercomparison between AMOR, miniDOAS and another set of passive samplers is obviously possible but is outside the scope of this paper.

(comment from referee)

Authors should provide R2 values for correlation plots in Figures 9 and 12.

(author's response)

We added R2 to the figure captions of Fig. 9 (page 28, line 6, hourly concentrations at Vredepeel, R2=0.70), Fig. 10 (page 29, line 2, hourly concentrations at De Zilk, R2=0.76) and Fig. 12 (page 30, line 2, daily concentrations at Vredepeel, R2=0.81).

**(comment from referee)**

I imagine that such a long intercomparison of co-located AMOR and DOAS instruments was partially conducted in order to aid future interpretation of long-term ammonia concentration trends. For this purpose, it would be also interesting to see a more in-depth analysis of the collected data. For example – the analysis of seasonal and geographical trends and/or differences in AMOR and DOAS datasets. Figure 13 shows monthly averages obtained by both instruments at all stations throughout the country. It is obvious that datasets for some stations agree better than for others. I found it disappointing that analysis was only limited to a correlation plot.

**(author's response)**

The aim of this paper is indeed to join the datasets of AMOR and miniDOAS, so that concentration trend analyses spanning the changeover from AMOR to miniDOAS are possible. In the future, the miniDOAS dataset will be highly valuable to do analyses per season or per station, on a higher temporal resolution than is possible with the AMOR data. The interpretation of the current dataset may change if more data becomes available and if we gain more insight in the performance of the instrument over time. For
now, a subdivision of the dataset seems not feasible, especially not the intercomparison between AMOR and miniDOAS data; per station, there are only 12 to 16 monthly data points available.

(comment from referee)

Authors should add s description of the expected maintenance schedule and expected lifetime for DOAS instruments and their major components for the DOAS instruments.

(author's response)

Thank you for this suggestion. We add a short section to the manuscript (page 5, line 12):

(author's change in manuscript)

2.3.2 Maintenance schedule and instrument lifetime

Based on the experience gained during this intercomparison, we now adhere to the following maintenance schedule:

- The xenon lamp is exchanged once a year. Because replacing the lamp makes measuring new reference spectra necessary (see Sect. 2.3.4), such a replacement takes place by exchanging a system with a life-expired lamp with a system with a new lamp installed and newly measured reference spectra present. This minimises the instrument downtime.

- About half a year after a system is installed at a station, it receives a service visit. The system is cleaned and the alignment of the optics is checked.

- The quartz window in the station housing is cleaned every four weeks.

- The parabolic sender mirror is replaced after 5 years of continuous use.

We estimate the lifetimes of the other optical components – the receiver mirror, the folding mirror and the interference filter – to be between 5 and 10 years. The electronic
parts – the alignment correction motors, the spectrograph and the instrument computer – also have lifetimes between 5 and 10 years. The remaining parts – the breadboard and the mirror mounts – have lifetimes of more than 25 years.

(comment from referee)

Minor comments Replace "Uptime" with "Data capture rate"

(author's response)

We propose to ask the English language editor for advise on this change.

(comment from referee)

Replace "life-expired" with outdated in reference to old instrumentation; and with "burned-out" in reference to burned-out LP-DOAS lamp

(author's response)

"Outdated" expresses the opinion that the instrument is no longer state-of-the-art, or even that it is old-fashioned and no longer suitable for use. We intended to denote that the instruments reached the end of their economical lifespan, with repairs becoming more frequent and more costly. We propose to ask the English language editor for advise on this change.

"burned-out": we replace the lamps preferably before they burn out, so this is probably not a good change to make.

(comment from referee)

Page 2, line 16 (and through the rest of the document) - abbreviations shall be presented in parenthesis when first used in the text, and not another way around. E.g. – Differential Optical Absorption Spectroscopy (DOAS).

(author's response)

We changed all instances, on page 2, line 16; page 3, line 11; page 4, line 7 and page
5, line 9.

(comment from referee)

Page 4 – combine sections 2.2.1 through 2.2.4 into single section 2.2.

(author's response)

Thank you for this suggestion, we merged these sections.

(comment from referee)

Page 6, Equation 6 - should be 2l in denominator.

(author's response)

In this equation, as in equations 1 to 5, I is the full optical length, i.e. twice the distance between the instrument and the retroreflector. We made this more explicit in the definition, on page 5 line 20:

(author's change in manuscript)

... I is the length of the full optical path. In a monostatic system like ours, I is twice the distance between the instrument and the retroreflector.

(comment from referee)

Page 10, line 28 - replace "overview of the situation" with "measurements setup".

(author's response)

We made this change.

(comment from referee)

Pages 13-15 – combine section 4.1 and 4.2, and 4.4. Pages 14-15 – eliminate subsection in 4.3 and make a more fluid narrative (e.g. section 4.3.1 can be reduced to a single sentence or illuminated completely as it was already discussed in section 3.2).
(author's response)

Thank you for the suggestion. We made both these changes. The revised Section 4 is included as a supplement to this comment.

(comment from referee)

Please provide explanation for describing the DOAS instruments deployed in the network as miniDOAS. What is the main feature that differentiates the miniDOAS from a "traditional" LP-DOAS instrument?

(author's response)

That is indeed not clear from this paper. We added a short explanation on page 5, line 11:

(author's change in manuscript)

It was developed from a much larger system, also described in Volten et al. (2012). By using smaller and less expensive parts, the physical dimensions, the power consumption and the price tag of the miniDOAS were much smaller than the original system, hence the name. See Sect. 2.3.4 and Volten et al. (2012) for an overview of the differences between the systems.

(comment from referee)

Table 1 can be augmented to include physical dimensions of the different instruments.

(author's response)

With the exception of the miniDOAS, no dimensions were published for any of the instruments listed in Table 1. This makes adding the dimensions to the Table not feasible. We agree that information on the size of the instrument can be useful, we therefore added a photograph of the instrument (Fig. 3, referred to on page 5, line 11), with the physical dimensions in the Figure caption. This Figure is included at the end
of this author's response.

Please also note the supplement to this comment: https://www.atmos-meas-tech-discuss.net/amt-2016-348/amt-2016-348-AC2supplement.pdf

**Supplement:**

**4    Discussion**

Analysing the results obtained in the comparison we see that the uptime of both instruments is comparable. At 80-90%, the miniDOAS uptime is adequate for an instrument in an automated monitoring network. We expect that the uptime of the miniDOAS will further improve in 2016, as from then on the instrument will benefit from the regular monitoring of the network performance.

**4.1    Timescale of the intercomparison**

Both instruments provide minute values of ambient ammonia concentrations. When looking at short timescales (minutes, hours) we see relatively large differences between the datasets. The differences get smaller as the timescale gets longer. When we look at the fits in the scatterplots of hourly, daily and monthly averages (Fig. 11, Fig. 12 and Fig. 16, respectively) we see that the slopes approach unity: 1.54 for hourly averages, 1.27 for daily and 1.03 for monthly averages. The offsets approach zero, from -7.34 µg m$^{-3}$ for hourly averages, via -3.06 µg m$^{-3}$ for daily to 0.65 µg m$^{-3}$ for monthly averages.

We conclude that, on a timescale of minutes or even hours, the instruments do not compare well. This is caused by a distinct difference in temporal resolution: the typical integration time of the miniDOAS is 1 minute, its minute-measurements are delay-free and mutually independent. The AMOR has a much larger response time, despite its claimed temporal resolution of 3 minutes. Its response to abrupt changes shows a delay (order of 30-60 minutes) and a spread-out and flattening of short peaks. In general, the integral over time of the AMOR-observed ammonia seems to remain conserved, as is reflected by the good comparison of longer timescale averages discussed above. This means that (virtually) no ammonia is lost in the AMOR, but it will be recorded at a different moment in time than its actual appearance at the AMOR inlet.

On timescales of hours, e.g. when looking at daily cycles, we consider the miniDOAS concentrations to be more representative for the actual ambient ammonia concentrations than the AMOR measurements.

We therefore focused our comparison on longer timescales: daily and monthly values. Daily value pairs showed good agreement in a direct comparison, i.e. when the concentration values are plotted in the same graph (see e.g. Fig. 13). The smoothing and delay effects that are apparent in the minute and hourly values have largely disappeared. However, scatter plots (see e.g. Fig. 14) show still some deviations from y = x, indicating that some delay effects are still not smoothed out. This is to be expected, a high peak just before the transition to a new day will cause differences in two consecutive days. In the monthly averages, because there are far fewer transitions, such extremes have disappeared. This makes monthly averages the timescale of choice for the intercomparison.

The monthly averaged concentrations show a linear relationship, as indicated above. We conclude that for monthly averages the instruments compare well. Over the whole comparison period there is an average offset of 0.65 ± 0.28 µg m$^{-3}$ and a slope of 1.034 ± 0.028 between the techniques. Thus, the miniDOAS measures on average slightly higher than the AMOR, over all concentration ranges.

From a scientific point of view this correspondence is excellent, especially since two completely different measurement techniques are used. As a reference, we refer to a study by Von Bobrutzki et al. (2010) that shows much larger discrepancies between different techniques. The systematic difference found between AMOR and miniDOAS amounts to roughly 10% of the typical ammonia background concentrations in the Netherlands of around 5 µg m$^{-3}$. Fortunately, not so much the absolute difference between the techniques is politically relevant but any jumps in ammonia trends, see e.g. van Zanten et al. (2017) or Wichink Kruit et al. (2017) for two studies in which this data is being used. We will discuss some possible explanations for the difference between the techniques.

**4.2    Possible explanations for the difference between the techniques**

A possible effect of the difference in measurement altitude (the AMOR measured at 3.5 m, the miniDOAS at 2.2 m) was studied using passive samplers. As discussed in Sect. 3.2, the results show no significant difference between AMOR inlet and miniDOAS path, so they offer no explanation for the observed positive bias between miniDOAS and AMOR. Further research with more precise equipment would be needed to reduce the statistical error in these measurements and study the effects of the altitude difference between 2.2 and 3.5 m, as well as the possible influence of the station housing.

The AMOR validation procedure may influence the intercomparison as well. AMOR data is validated based on concentration data only. No AMOR instrument parameters are included in the validation procedure. Closer inspection of validated AMOR values show periods after maintenance where values approved in the validation procedure may - in hindsight - be considered too low and erroneous. This conclusion can only be drawn when the miniDOAS dataset is used as an additional validation tool. Removing these data from the comparison (dubious as it would be from a scientific point of view) would however improve the comparison only slightly. Therefore, the validation procedure can be ruled out as a major source of the offset.

Another effect to be looked at is a possible loss of ammonia in the AMOR air inlet system, as this is a known effect in ammonia inlet lines (Yokelson et al., 2003). However, the AMOR air inlet system has been designed to minimise such effects. Especially the relatively high airflow through the instrument, of 25 L min$^{-1}$ rather than the mL min$^{-1}$ flows found in other instruments, should be effective in minimising these effects. As discussed in Sect. 3, no indication for ammonia loss was found in the measurement data. It seems therefore unlikely that ammonia loss is a major contributor to the bias found.

The AMOR calibration procedure should come up for scrutiny as well. AMOR calibrations are performed using calibration fluids, and thus only pertain to the 'liquid' part of the instrument, after ammonia has been absorbed in the denuder. Any losses in the airborne phase, e.g. in the inlet system, are not included in the calibration procedure. As stated previously, the reason for omitting this part in the calibration procedure is that it is virtually impossible to generate an adequate calibrated gas flow, as the AMOR tries to minimize inlet effects by using a very high airflow of 250 m$^3$ h$^{-1}$, from which a further 25 L min$^{-1}$ is sampled by the instrument. We have not been able to study this aspect further in the framework of this comparison.

16

Shifting our attention to the miniDOAS, we note that the miniDOAS zero is determined by comparison to a DOAS reference instrument. Any offset in the reference instrument will show up as a similar offset in the reported miniDOAS values. The zero of the reference instrument is determined by study of a long time series, looking for periods of lowest values and assuming these occur at constant zero ammonia levels. If this assumption is incorrect, it results in the reference instrument underestimating the real concentrations. This would therefore lead to a negative bias in the concentrations reported by the miniDOAS, never to a positive one. There is no evidence for this in the dataset.

17

[Figure]

**Figure 5: Spectral fit and concentration retrieval shown for a typical measurement. Left from top to bottom: the measured spectrum $I_{meas}(\lambda)$ (1 minute average); the background corrected spectrum $I_{bgc}(\lambda) = ( I_{meas}(\lambda) - I_{dark}(\lambda) ) / I_{background}(\lambda)$ (see Eq. 7); the DOAS curve $DC(\lambda) = \ln( I_{bgc}(\lambda) / [I_{bgc}(\lambda)]_{moving\ average} )$ (see Eq. 4), the fit to the DOAS curve $fit(\lambda) = \alpha \cdot X(\lambda) + \beta \cdot Y(\lambda) + \gamma \cdot Z(\lambda)$ (see Eq. 5) and the residual $residual(\lambda) = DC(\lambda) - fit(\lambda)$. Right from top to bottom: NH₃, SO₂ and NO reference spectra used in the fitting procedure. Scales for the y-axes are arbitrary. Units for the y-axes of the reference spectra are the same. The residual is a tool that helps to identify potential flaws in the measurements, such as interference problems. The dark grey lines in the bottoms of the graphs show the analysis interval used. The measurement shown here was taken at Vredepeel on 23 January 2015, 6:36 UTC.**